# Exercise induces cerebral VEGF and angiogenesis via the lactate receptor HCAR1

Cecilie Morland[1,2,3,*], Krister A. Andersson[1,2,3,*], Øyvind P. Haugen[1], Alena Hadzic[1,2,3], Liv Kleppa[1,3], Andreas Gille[4], Johanne E. Rinholm[1,3], Vuk Palibrk[5], Elisabeth H. Diget[1,6], Lauritz H. Kennedy[1,3], Tomas Stølen[7], Eivind Hennestad[8], Olve Moldestad[9], Yiqing Cai[1], Maja Puchades[3], Stefan Offermanns[10], Koen Vervaeke[8], Magnar Bjørås[5], Ulrik Wisløff[7], Jon Storm-Mathisen[3] & Linda H. Bergersen[1,3,6]

Physical exercise can improve brain function and delay neurodegeneration; however, the initial signal from muscle to brain is unknown. Here we show that the lactate receptor (HCAR1) is highly enriched in pial fibroblast-like cells that line the vessels supplying blood to the brain, and in pericyte-like cells along intracerebral microvessels. Activation of HCAR1 enhances cerebral vascular endothelial growth factor A (VEGFA) and cerebral angiogenesis. High-intensity interval exercise (5 days weekly for 7 weeks), as well as L-lactate subcutaneous injection that leads to an increase in blood lactate levels similar to exercise, increases brain VEGFA protein and capillary density in wild-type mice, but not in knockout mice lacking HCAR1. In contrast, skeletal muscle shows no vascular HCAR1 expression and no HCAR1-dependent change in vascularization induced by exercise or lactate. Thus, we demonstrate that a substance released by exercising skeletal muscle induces supportive effects in brain through an identified receptor.

[1] The Brain and Muscle Energy Group, Electron Microscopy Laboratory, Department of Oral Biology, University of Oslo, NO-0316 Oslo, Norway. [2] Institute for Behavioral Sciences, Faculty of Health Sciences, Oslo and Akershus University College, NO-0167 Oslo, Norway. [3] The Synaptic Neurochemistry Lab, Division of Anatomy, Department of Molecular Medicine, Institute of Basic Medical Sciences, Healthy Brain Ageing Centre, University of Oslo, NO-0317 Oslo, Norway. [4] Institute for Experimental and Clinical Pharmacology and Toxicology, Mannheim Medical Faculty, Heidelberg University, D-68169 Mannheim, Germany. [5] Department of Cancer Research and Molecular Medicine, Norwegian University of Science and Technology, NO-7491 Trondheim, Norway. [6] Center for Healthy Aging, Department of Neuroscience and Pharmacology, Faculty of Health Sciences, University of Copenhagen, DK-2200 Copenhagen N, Denmark. [7] K.G. Jebsen Center of Exercise in Medicine, Department of Circulation and Medical Imaging, Norwegian University of Science and Technology, NO-7491 Trondheim, Norway. [8] Laboratory of Neural Computation, Department of Physiology, University of Oslo, NO-0317 Oslo, Norway. [9] Centre for Rare Disorders, Oslo University Hospital, Rikshospitalet, NO-0424 Oslo, Norway. [10] Max-Planck-Institute for Heart and Lung Research, Department of Pharmacology, D-61231 Bad Nauheim, Germany. * These authors contributed equally to this work. Correspondence and requests for materials should be addressed to C.M. (email: cecilie.morland@hioa.no) or to L.H.B. (email: l.h.bergersen@odont.uio.no).

Exercise has beneficial effects on the brain, which is especially important in the elderly[1–3]. Increased density of capillaries due to angiogenesis, the sprouting of new capillaries from pre-existing vessels, is one mechanism through which exercise improves brain function[4]. In fact, a positive correlation between cognition and cerebral perfusion has been demonstrated in several studies[5]. Increased vascular density in the brain in response to exercise may therefore be particularly important to maintain cognitive performance during normal ageing, age-related dementias (including the most prevalent: vascular dementia and Alzheimer's disease) and Parkinson's disease, as well as in protection against ischaemia. All of the mentioned conditions are associated with reduced metabolic capacity and reduced density of microvessels in the brain, paralleled by chronic cerebral hypoperfusion[6–8]. Together, these features may contribute to declining cognitive functions such as seen in the elderly. Therefore, there is reason to believe that some of the positive effects of regular physical exercise on the brain are direct consequences of enhanced cerebral perfusion through angiogenesis[9].

Angiogenesis is stimulated by vascular endothelial growth factor A (VEGFA)[10], which also directly enhances neurogenesis and synaptic function[11]; however, the initial molecular signal that leads to increased cerebral VEGFA in response to exercise has not been determined. Exercise at high intensity, causing lactate from active skeletal muscles to accumulate in the blood, and lactate injections have previously been found to increase brain expression of VEGFA[12], but the mechanism is unknown. Moreover, in wounds, lactate is known to accumulate and stimulate angiogenesis, but precisely how lactate acts has not been determined[13,14]. Lactate, released in situ from polymeric lactic acid microfibres, induces angiogenesis in the brain, again through unidentified mechanisms[15]. Cerebral hypoxia, another condition known to increase lactate levels in the brain, also causes angiogenesis via VEGFA[16]. However, as lactate or exercise does not increase hypoxia-inducible factor 1α (HIF-1α), hypoxia is unlikely to be part of the response[12]. The mechanisms behind lactate-induced angiogenesis thus remain to be elucidated. As L-lactate levels can increase by more than an order of magnitude during strenuous exercise[17], we hypothesized that the lactate receptor, hydroxycarboxylic acid receptor 1 (HCAR1, also known as HCA1 or GPR81), could mediate the signal. We recently discovered the lactate receptor HCAR1 to be present and active in the brain, downregulating cAMP[18].

Here we uncover that HCAR1 is highly enriched in leptomeningeal fibroblast-like cells that line and surround the pial blood vessels supplying the brain and also in pericyte-like cells on intracerebral microvessels, and that activation of this receptor stimulates cerebral VEGFA levels and angiogenesis, providing an initial link between exercise and brain sustenance.

## Results and Discussion

**HCAR1 mediates exercise-induced brain vascularization.** To investigate whether activation of HCAR1 could be an initial event leading to angiogenesis, we exposed wild-type mice and Hcar1 knockout mice[19] to high-intensity interval exercise 5 days a week for 7 weeks. This exercise regime has been developed to achieve optimum enhancement of cardiovascular function[20] and gives peak lactate levels of ∼10 mM (see Methods). After 7 weeks of exercise, wild-type mice, but not knockout mice, had an increased density of capillaries in the sensorimotor cortex (Fig. 1a,b), and in the hilus of the dentate gyrus of hippocampus (Fig. 1c,d), compared to sedentary controls. There was no change in the capillary diameter (Fig. 1d), indicating that the observed increase in vascular density reflects an increase in the area of contact

between blood and the brain. This effect did not reflect differences in running speed or exercise intensity, as wild-type and knockout mice showed equal performance levels in maximal exercise-capacity tests, which were performed every second week throughout the intervention period (Supplementary Fig. 1). As predicted with high-intensity exercise, the mice rapidly increased their fitness level during the first weeks of exercise, with the maximum running speed levelling off at 50% increase by week 5.

**Lactate injections increase brain angiogenesis.** Interestingly, the increased vascularization in wild-type mice, and the lack of effect in Hcar1 knockouts, was reproduced by daily subcutaneous injections of sodium L-lactate (2 g kg$^{-1}$ bodyweight; 200 mg ml$^{-1}$; pH 7.4; that is, 18 mmol kg$^{-1}$; raising blood lactate to ∼10 mM; Supplementary Fig. 2) 5 days a week for 7 weeks (Fig. 1). As lactate injections have been observed to cause acute anxiety attacks in susceptible individuals[21], the mice were tested 15 min after the first lactate injection in a rodent test for anxiety, the 'elevated zero maze' (see Methods), to rule out the possibility that the results were confounded by anxiety; the test showed no difference between the groups (Supplementary Fig. 3). We therefore conclude that lactate acting on HCAR1 is a pivotal regulator of angiogenesis in the brain and underlies the angiogenic effect of exercise.

**Angiogenic effect is large in the dentate gyrus.** The changes were relatively larger in the hilus of the dentate gyrus of hippocampus than in the sensorimotor cortex. The possibility therefore exists that the angiogenic effect of exercise and lactate is stronger in the dentate hilus, that is, a site known to undergo adult neurogenesis in response to exercise[22]. In the cerebellar cortex (Supplementary Fig. 4) there was no change, suggesting that these effects, observed in regions of the cerebral cortex, are not a general phenomenon of the brain.

**Exercise and lactate increase VEGFA levels in the hippocampus.** Angiogenesis in the developing nervous system[10] as well as in response to exercise[4] is regulated by VEGFA. We therefore measured the levels of VEGFA in wild-type and Hcar1 knockout mice after exercise or L-lactate treatment. We found increased VEGFA levels in the hippocampus of wild-type mice after exercise or L-lactate treatment compared to wild-type controls (Fig. 1e,g). In knockout mice neither of the treatments increased VEGFA levels above baseline (Fig. 1f,g). This indicates a direct link between HCAR1 activation and VEGFA signalling, resulting in angiogenesis. No similar changes in VEGFA were observed in the cerebellum (Supplementary Fig. 5). The uncropped VEGFA immunoblots are shown in Supplementary Fig. 6.)

**Perivascular pial and pericyte-like cells express HCAR1.** If VEGFA production occurs downstream of HCAR1, through activation of intracellular signalling pathways, then localization of HCAR1 in the vicinity of the blood vessels supplying the tissue is predicted. The Allen Brain Atlas (http://mouse.brain-map.org/gene/show/89056; http://mouse.b-rain-map.org/experiment/siv?id=77464856&imageId=77469798-&initImage=expression&contrast=0.5,0.5,0,255,4) indicates the most intense expression of Hcar1 mRNA in the pia mater, including meningeal blood vessels, in addition to much less-intense expression in principal cells (that is, pyramidal and granule cells) in mouse brain. Localization of HCAR1 protein at the blood–brain barrier, as well as in neurons and astrocytes has been reported by us, based on immunohistochemistry[18]. As a method alternative to immunocytochemistry, we here use transgenic mice[19] that express monomeric red fluorescent protein

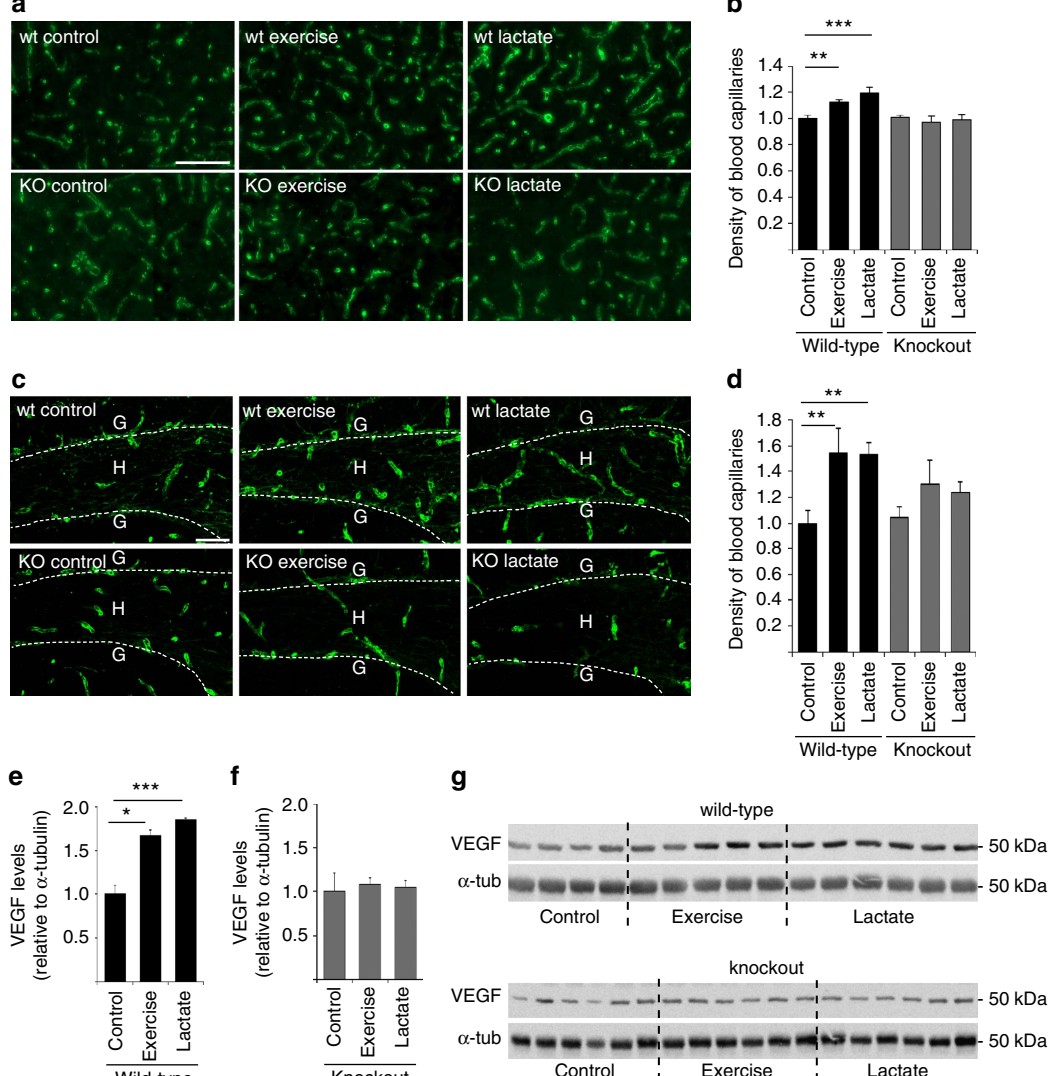

**Figure 1 | HCAR1 regulates VEGFA and capillary density in response to exercise.** (**a**) Collagen IV-labelled capillaries in the sensorimotor cortex grey matter of wild-type or *Hcar1* knockout mice exposed to vehicle injections (control), treadmill exercise or lactate injections, 5 days a week for 7 consecutive weeks. Scale bar, 100 µm. (**b**) Capillary density (per cent of the total area, normalized to wild-type control) in the sensorimotor cortex. Mean ± s.e.m. of $n = 7$ wild-type controls, seven wild-type exercise, six wild-type lactate, five knockout controls, four knockout exercise and six knockout lactate mice. Analysis of variance (ANOVA), $P = 0.001$; Fisher's least significant difference (LSD) *post hoc* test, **$P < 0.01$; ***$P < 0.001$. (**c**) Collagen IV-labelled capillaries in the dentate gyrus (DG) of the hippocampus of wild-type or *Hcar1* knockout mice treated as in **a**. Stippled line, the inner border of the granule cell layer (G), circumscribing the sampled area, hilus (H). Scale bar, 50 µm. (**d**) Capillary density (see **b**) in the hilus. Mean ± s.e.m. of $n$ mice ($n = 4-7$, as specified below for diameters). ANOVA, $P = 0.022$; Fisher's LSD *post hoc* test, **$P < 0.01$. Capillary external diameters (µm) in the same areas were unchanged (mean ± s.e.m. ($n$)): wild-type control 5.8 ± 0.2 (6), wild-type exercise 5.7 ± 0.1 (7), wild-type lactate 5.7 ± 0.2 (6), knockout control 5.8 ± 0.3 (5), knockout exercise 6.0 ± 0.1 (4), knockout lactate 5.8 ± 0.2 (6). ANOVA, $P = 0.93$. (**e**) Quantification of VEGFA in hippocampus of wild-type animals. Data are relative to α-tubulin (α-tub), normalized to wild-type control, mean ± s.e.m. of $n = 4$ wild-type control mice, five wild-type exercised mice and six mice treated with lactate, ANOVA, $P = 0.001$; Dunnetts's T3 *post hoc* test, *$P < 0.05$, ***$P < 0.001$. (**f**) Quantification of VEGFA in hippocampus of knockout animals presented as in **e**. $n = 6$ mice per group. ANOVA, $P = 0.88$. (**g**) Western blots of VEGFA underlying **e**,**f** (uncropped scans shown in Supplementary Fig. 6a).

(mRFP) under the *Hcar1* promoter (Supplementary Fig. 7b). This reporter protein spreads in the cytoplasm of cells that express endogenous HCAR1, but is not targeted to the surface membrane.

Surface views in an epifluorescence dissection microscope of fresh and freshly perfusion-fixed mouse tissues showed intense mRFP-HCAR1 labelling along pial blood vessels (Fig. 2a,b). Except for adipose tissue, HCAR1 labelling in other organs was hardly discernible above background. Furthermore, quantitative PCR (qPCR) data showed that the pia expresses higher levels of

*Hcar1* mRNA than does any other tissue examined, except fat (Supplementary Table 1). Surface view of the brains of mice killed without fixation revealed that the labelled vessels included branches of the medial cerebral artery as well as veins that drain into the sagittal sinus (Fig. 2a,b). Two-photon imaging *in vivo* showed that mRFP-HCAR1 fluorescent cells are spread over the pia mater, enriched close to pial blood vessels (Fig. 2c,d). Fluorescence angiography after intravascular injection of fluorescein isothiocyanate (FITC)–dextran (Fig. 2e) showed a fluorescence-weak part of the wall between the vessel lumen

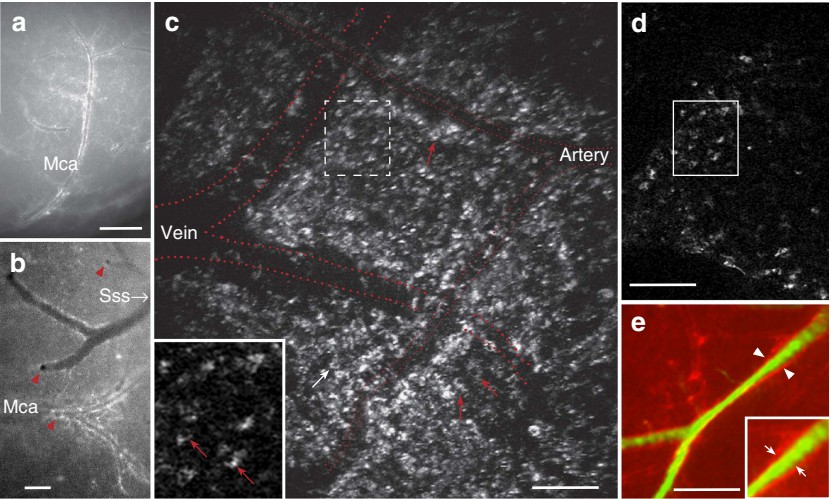

**Figure 2 | Survey of HCAR1 in pia mater and at pial blood vessels supplying the brain. (a,b)** Surface views of the brain of a mRFP-HCAR1 reporter mouse. **(a)** Right arteria cerebri media (Mca), lateral view. Scale bar, 900 μm. **(b)** Branch of left Mca, top view, and vein draining into the superior sinus sagittalis (Sss). Branches penetrating the brain parenchyma are seen as black holes in **b** (red arrowheads). mRFP fluorescence appears enriched along blood vessels. Scale bar, 300 μm. **(c,d)** Two-photon imaging of pia mater in a live and walking mRFP-HCAR1 reporter mouse showing leptomeningeal cells (arrows) in the vicinity of two pial blood vessels (approximate vessel positions indicated by red dotted lines). **(c)** Stack of superimposed optical sections. Scale bars, 100 μm. **(d)** Single optical section (1 μm); frame shown is magnified as inset in **c**; stippled frame in **c** indicates) position of frame in stack. **(e)** Surface view of the cortex from mRFP-HCAR1 reporter mouse showing fluorescent pial vessel outline (arrowheads) and lumen (green) after retrograde injection of FITC-dextran agarose in the v. jugularis. Scale bar, 300 μm. Inset: magnified view showing space between green and red corresponding to vessel wall (small arrows).

and the surrounding mRFP-HCAR1, suggesting that the HCAR1-bearing cells did not directly contact the blood stream. In sections of fixed brain (Fig. 3a–l), the fluorescence of mRFP was boosted by mRFP-selective antibodies (Supplementary Fig. 7). Co-staining with the basal lamina marker collagen IV (Fig. 3a,b,i) showed that the mRFP-HCAR1-labelled pial cells are situated on and near the blood vessel wall. They co-localize with the fibroblast/mesenchymal cell marker vimentin (Fig. 3b), in agreement with the fibroblast nature of leptomeningeal cells of the pia mater. In sections perpendicular to the brain surface, the pia was seen as an mRFP-HCAR1 fluorescent monolayer (Fig. 3c). The blood vessels penetrating into the brain parenchyma had less-intense and less-regular labelling (Fig. 3d), but a sheath of mRFP-HCAR1 fluorescent cells accompanied some of the vessels (Fig. 3d–l and Supplementary Fig. 8a), gradually vanishing with distance from the pial surface. No fluorescence along such vessels was present in non mRFP mice (Supplementary Fig. 8b).

4′,6-Diamidino-2-phenylindole dihydrochloride (DAPI) staining of nuclei indicated that the mRFP-HCAR1 fluorescent cells are located abluminally of the endothelium (Fig. 3h,j–l). Co-labelling with markers of endothelial surface (platelet endothelial cell adhesion molecule, also known as cluster of differentiation 31, CD31) showed that the mRFP-HCAR1 fluorescent sheath is clearly separate from and surrounds the endothelial cells (Fig. 3e–h), but did not exclude a slight mRFP-HCAR1 signal in the endothelium. All of the mRFP-HCAR1 fluorescent cells along intracerebral blood vessels appear to co-localize with platelet-derived growth factor receptor β (PDGFRβ), a pericyte-associated protein[23], suggesting that these are pericyte-like cells (Fig. 3j–l). Like leptomeningeal cells, pericytes are akin to fibroblasts and myoblasts and are known to be involved in the tuning of capillary vasomotor function[24] as well as angiogenesis[25,26], involving, *inter alia*, PDGFRβ signalling[25]. Lactate, but not HCAR1, has previously been implicated in the control of pericyte function[27].

Immunocytochemical labelling for HCAR1 was performed on paraffin sections with a commercial antibody, after absorption of the antibody with brain sections from *Hcar1* knockout mice in order to remove antibodies reacting with other proteins than HCAR1 (Fig. 4). Immunolabelling of leptomeningeal cells in pia mater and associated with blood vessels was observed in wild-type (Fig. 4a), but not in knockout tissue (Fig. 4b). HCAR1-immunolabelled perivascular cells that also contained mRFP were observed in tissue from mRFP-HCAR1 reporter mice (Fig. 4c,d,e). This directly demonstrates HCAR1 in the leptomeningeal cells and confirms that the mRFP reporter-expressing cells, as expected, also express the endogenous HCAR1.

The intimate adjacency of the HCAR1-carrying cells to the blood vessels supplying the brain parenchyma implies a particular significance in relation to peri-/paravascular drainage of brain interstitial fluid, which is currently a focus of high interest[28,29]. (Note the subpial/paravascular/perivascular space in Fig. 3f,h ∼ 'Virchow–Robin space' (also see Fig. 3 of ref. 28 and Fig. 3 of ref. 29)). Thus, HCAR1 has an ideal localization (Fig. 5), at the blood–brain interface, to detect fluctuations in brain interstitial fluid lactate when the metabolic state of the brain changes, and also to detect increases in circulating lactate during exercise. The reported $EC_{50}$ for the receptor HCAR1 is in the low millimolar range, reaching saturation at $\sim 30$ mM (refs 30,31). Therefore, HCAR1 can respond to lactate within the full concentration range reported *in vivo* (0.5–30 mM in blood and brain extracellular fluid, see review ref. 32).

**No HCAR1-dependent angiogenesis in peripheral organs.** VEGFA synthesis and release are known to be regulated by several parallel signalling pathways, such as the MAP kinase and PI3 kinase/Akt pathways, both leading to an increase in HIF-1α (ref. 33), but also by HIF-1α-independent pathways, including through PPARγ coactivator 1-α (refs 12,34). Regulation of these

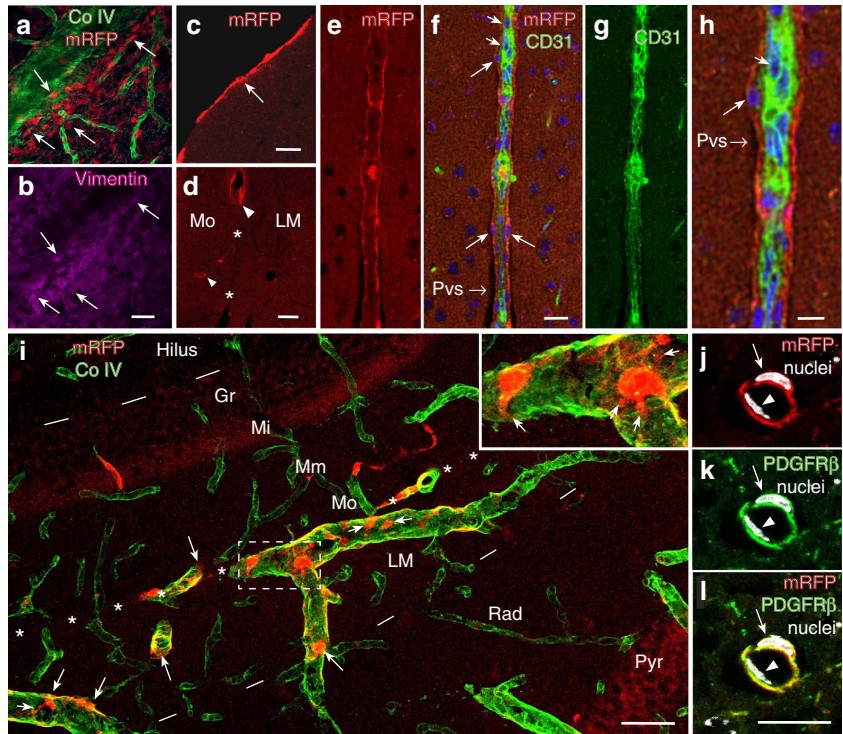

**Figure 3 | Details of HCAR1-expressing cells in pia mater and pial and brain vessels.** (**a,b**) Pial vessel with emerging capillaries labelled for basement membrane collagen IV (CoIV, green), decorated by leptomeningeal cells that contain mRFP-HCAR1 (red) and co-localize fibroblast marker vimentin (magenta, **b**); confocal images from top frozen section tangential to the brain surface. Scale bar, 70 μm. (**c**) Pial cells (arrow) in confocal image of parasagittal frozen section perpendicular to the cerebral cortex of mRFP-HCAR1 reporter mouse. Scale bar, 20 μm. (**d**) In the extension of pia in fissura hippocampi (*), which separates hippocampus stratum lacunosum-moleculare (LM) from the molecular layer of area dentata (Mo), mRFP-HCAR1 is in select blood vessels (arrowheads), including ones (small arrowhead) penetrating into the Mo. Scale bar, 50 μm. (**e–h**) Blood vessel penetrating into the cerebral cortex with a sheath of mRFP-HCAR1-containing perivascular cells (long arrows), CD31 in endothelial cells (short arrows, green; **f–g**) and DAPI-stained cell nuclei (blue; **f**); single optical section (1.38 μm). The picture is compatible with a slight mRFP-HCAR1 signal also in endothelial cells. Subpial/paravascular/perivascular space (Pvs). Scale bar, 20 μm (**e–g**). (**h**) Magnification of part of **f**. Scale bar, 10 μm. (**i**) mRFP-HCAR1 in the hippocampus, magnified frame showing details of mRFP-HCAR1-carrying cells (arrows, red), which extend processes (small arrows) around blood vessels (CoIV-labelled, green) that penetrate into the hippocampus through the extension of pia in the fissura hippocampi (indicated by white asterisks, brain surface to the left outside the picture). Hil, hilus of area dentata; Gr, granule layer of area dentata; Mi, Mm, Mo, inner, middle and outer zones of the molecular layer of area dentata; LM, Rad and Pyr, lacunosum-moleculare, radiatum and pyramidal layers of hippocampus CA1. Dashes mark borders between Hil and Gr, and between LM and Rad. Scale bar, 50 μm. (**j–l**) Small vessel in the cerebral neocortex, surrounded by cell co-expressing mRFP-HCAR1 (**j**, red; **l**, yellow) and the pericyte-associated protein PDGFRβ (**k**, green; **l**, yellow). Staining of nuclei (arrow and arrowhead, DAPI, white) reveals that an endothelial cell (arrowhead) is located between the lumen and the mRFP-HCAR1/PDGFRβ-co-expressing cell. Scale bar, 15 μm.

pathways by growth factor receptors has been described[33]. Regulation of VEGFA by circulating insulin-like growth factor 1 (IGF-1) has been suggested[35]; however, in the present study we did not find evidence for *Hcar1* mRNA expression in the liver (Supplementary Table 1), the main organ responsible for release of IGF-1 to the circulation[36].

If an increase in circulating growth factors such as IGF-1 were responsible for the increased cerebral vascularization, the same effect should be seen in other organs that express growth factor receptors, such as skeletal and heart muscle. We therefore investigated the effects of exercise and lactate injections on vascularization in skeletal muscle. Although low levels of *Hcar1* mRNA (Supplementary Table 1) and HCAR1 protein have previously been reported in skeletal muscle[31], we found no significant level of the reporter protein in muscles of mRFP-HCAR1 reporter mice (Supplementary Fig. 9). Accordingly, we did not find HCAR1-dependent enhancement of vascularization in skeletal muscle (Supplementary Fig. 10). Therefore, we conclude that the increases in VEGFA levels and cerebral angiogenesis reported here probably occur downstream of the activation of HCAR1 in the pia and/or brain.

**HCAR1 stimulation activates ERK1/2 and Akt signalling.** To characterize the mechanisms linking HCAR1 activation to VEGFA at the cellular level, we performed *in vitro* experiments (Supplementary Fig. 11) with hippocampal slices from knockout and wild-type mice. HCAR1 could act through several mechanisms and could theoretically regulate the same pathways that are known targets for growth factor receptors. HCAR1 activates extracellular signal-regulated kinase (ERK1/2) through pertussis toxin-sensitive pathways, independently of arrestin[37]. In addition, HCAR1 can act through arrestin-β2, to protect against inflammasome-mediated cell damage[38]. Importantly, activation of phosphatidylinositol 3-kinase/Akt (PI3K/Akt) and ERK1/2 has been found to mediate increased expression and secretion of VEGFA[39]. We show that stimulation by lactate or the selective HCAR1 agonist 3,5-dihydroxybenzoate caused phosphorylation of ERK1/2 as well as Akt in hippocampal slices from wild-type but not *Hcar1* knockout mice (Supplementary Fig. 11a–d), consistent with activation of Akt and ERK1/2 being mediators of our observed effect of HCAR1 on VEGFA and angiogenesis in hippocampus.

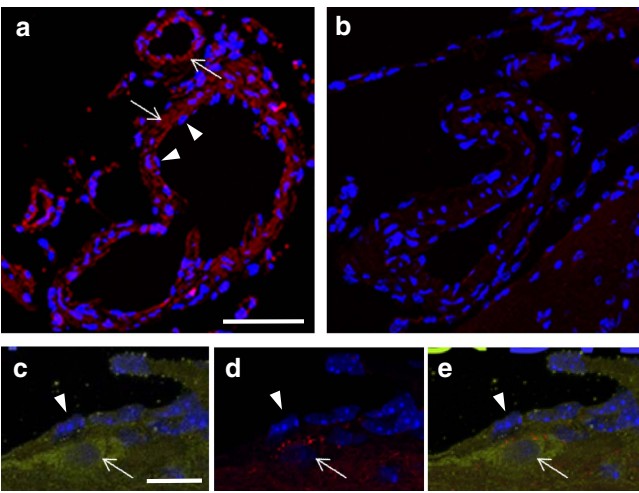

**Figure 4 | HCAR1 immunoreactivity colocalizes with mRFP-HCAR1 at pial vessels.** (**a,b**) HCAR1 immunoreactivity (red) is in the wall of blood vessels in pia mater (arrows), in wild-type (**a**), but not in knockout (**b**). Nuclei of endothelial cells (arrowheads) are on the luminal side of the immunoreactivity (**a**). Scale bar, 30 μm. (**c–e**) HCAR1 immunoreactivity (green, **c,e**) is highly expressed in a leptomeningeal cell (arrow) on the abluminal side of the endothelium (arrowhead), which shows a slight HCAR1 immunoreactivity. In this reporter mouse, immunoreactivity to mRFP-HCAR1 (red, **d,e**) is seen in cytoplasmic granules in the leptomeningeal cell. Nuclei are blue (DAPI). All sections are from paraffin-embedded brain tissue and exposed to heat-induced epitope retrieval. Scale bars, (**a,b**), 30 μm; (**c–e**), 10 μm.

We have identified the lactate receptor HCAR1 as a key regulator of VEGF and angiogenesis in the brain (Fig. 4) and as an initial mediator of cerebral effects of physical exercise. Since brain dysfunctions, including ageing and age-related dementias, are associated with chronic hypoperfusion and microvascular dysfunction[6–8], our findings pinpoint enhanced HCAR1 activation as a potential new therapeutic strategy for treatment against cognitive decline and other brain conditions associated with hypoperfusion and energy deficiency. A potential 'exercise pill'[40] targeting HCAR1 may be useful to boost (but not replace) the effects of physical exercise, particularly in people at risk of developing dementia, who are typically unable to achieve high exercise levels.

## Methods
**Animals.** Animals used in this study were treated in strict accordance with the national and regional ethical guidelines. All experiments were performed by FELASA-certified personnel and approved by the Animal Use and Care Committee of the Institute of Basic Medical Sciences, The Faculty of Medicine, University of Oslo, and by the Norwegian Animal Research Authority (FOTS6292, FOTS6505, FOTS6590, FOTS6720, FOTS6758 and FOTS8243). The generation of the *Hcar1* knockout line and the mRFP-HCAR1 reporter mouse line has been described[19]. Specifically, the open reading frame of the mouse *Hcar1* gene carried by the 197 kb large BAC RP23-91D24 (CHORI) was replaced by a cassette containing the mRFP, as described previously[41]. Both the knockout line and the mRFP line were maintained in C57Bl/6N background in Bad Neuheim and in Oslo. Genotypes were verified by Southern blotting in the mice to be used in experiments (Supplementary Fig. 7).

**Animal treatment.** All mice were 7–9 weeks of age at the start of the experiments. *Hcar1* knockout or wild-type mice (both sexes with about equal M/F distribution, see Supplementary Fig. 1) were randomized into three groups: treadmill running, sodium L-lactate injections or saline injections (control). The mice that were treated with lactate received a subcutaneous injection of sodium L-lactate (≥99.0%, Aldrich, 71718; 2 g kg$^{-1}$ bodyweight; 200 mg ml$^{-1}$ dissolved in 0.9% saline; pH-adjusted to 7.4; that is, 18 mmol kg$^{-1}$). The control mice received the same volume (per kg bodyweight) of 0.9% saline. The lactate or saline injections

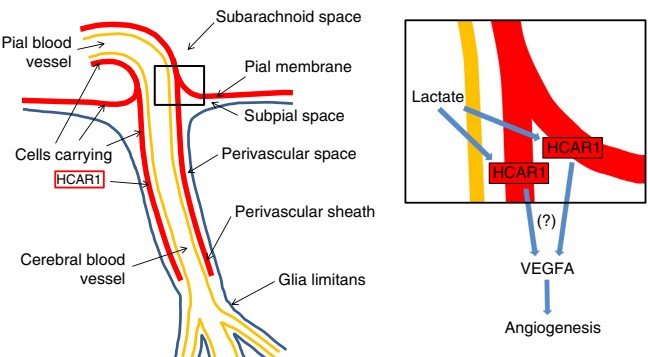

**Figure 5 | Organization of cells that carry HCAR1 and of the angiogenic action of lactate.** Blood-borne lactate from exercising muscle penetrates the blood vessel wall (yellow) through monocarboxylate transporters located in the vascular endothelium[53] (which represents the blood–brain barrier). Extravascular lactate (from blood or generated in the brain parenchyma upon neural activation) is freely diffusible in the perivascular/subpial space, thereby bathing the leptomeningeal fibroblast-like cells carrying HCAR1 (red). Magnified inset indicates possible, yet unidentified (?), pathways leading from activation of HCAR1 in the cells in pia and perivascular sheaths to increased VEGFA and subsequent enhanced angiogenesis. The perivascular sheath extends as separate HCAR1-expressing pericyte-like cells at intracerebral microvessels, which may also contribute in the angiogenic process. Although apparently devoid of mRFP-HCAR1, other cells may possibly express low levels of HCAR1. HCAR1 may stimulate VEGFA in the same cells, or in other cells, through mediators. In addition to its angiogenic action, VEGFA has neurotrophic effects[11]. Importantly, all blood to the brain parenchyma has to pass in close proximity to the perivascular sheath of HCAR1-carrying cells and therefore can convey products released from these cells upon activation of the receptor; blood to the hippocampus passes through vessels (such as the ones shown in Fig. 3i) entering in the hippocampal fissure, an extension of the pia mater that penetrates deep into the centre of the hippocampal formation. (The anatomical sketch is based on ref. 54).

were administered subcutaneously 5 days a week for 7 weeks. The mice were weighed every week for adjustment of the dose.

Since lactate injections have been associated with acute anxiety attacks in susceptible individuals[21], the mice were tested at 15 min after the first lactate injection (Supplementary Fig. 3) by the elevated zero maze, an established rodent test of anxiety[42]. The mice were placed on a circular white track, diameter 60 cm, width 5 cm, elevated 60 cm above the floor, with two quarters being open and two quarters closed by walls. The mouse was placed with the head facing a closed zone and allowed to explore the maze for 5 min. Their movements were recorded with a tracking camera connected to a computer with the ANY-maze Video Tracking Software (Stoelting Co., USA). The % of time spent in the closed zone and the frequency of head-dips in the open zone were recorded as positive and negative correlates of anxiety, respectively. Between the individual mice, the track was carefully wiped off (with a cloth moistened with 70% ethanol) and dried to avoid interference from the smell of other mice[43].

The interval exercise regime has been previously described[20,44]. It is designed for optimum gain in cardiovascular function and to reach ~90% of VO$_{2max}$ during the high-intensity intervals. Briefly, each session consisted of 10 min warm-up at 5 m min$^{-1}$, followed by 10 high-intensity intervals of 4 min each, and separated by 2 min of active rest. Running took place on a treadmill (Columbus Instruments, USA) at a 25 degrees incline. The mice were exposed to the high-intensity interval exercise protocol for 5 consecutive days each week, for a total duration of 7 weeks. On the first day of the exercise intervention, and then every other week (Supplementary Fig. 1), a maximal exercise-capacity test was performed for each individual mouse, to adjust the running speed of the training intervals to near the maximum they could sustain during 10 consecutive intervals. After a 15 min warm-up period at 9.6 m min$^{-1}$, the band speed was increased by 1.8 m min$^{-1}$ every 2 min until exhaustion, that is, the mice refused to run further, despite being manually placed back on to the band or receiving electrical stimuli (maximally 1–2 per day by the intrinsic device of the treadmill). During the 7 weeks, the maximal exercise-capacity test results, measured as maximum running speed (highly correlated with VO$_{2max}$), levelled off at a 50% increase in capacity (Supplementary Fig. 1), showing that the training sessions were of intended intensity[45]. This exercise regime has been validated extensively[46].

Blood lactate levels at $\sim 10\,mmol\,l^{-1}$ have been reported in mice during treadmill exercise at close to $VO_{2max}$ (Desai and Bernstein[47], Fig. 4) and at close to maximum speed[19].

**Plasma lactate measurements.** For plasma lactate measurements, the mice were anaesthetized mildly with isoflurane and decapitated at exactly 5, 15, 30, 60 or 180 min after the dose of lactate ($n = 5$ for each time point) or saline ($n = 3$ for each time point). The blood was collected into 0.5 ml Minicollect tubes (Greiner Bio-One GmbH, Kremsmünster, Austria) containing 2.5 mg sodium fluoride (NaF) and 2.0 mg potassium oxalate (KOx) per ml to prevent *post vivo* glycolysis (with the formation of lactate) and coagulation, respectively[48]. Blood samples were centrifuged, and the resulting plasma samples were frozen in liquid nitrogen and stored at $-80\,°C$. The lactate measurements were performed using VITROS DT60II and VITROS LAC DT slides (Ortho Clinical Diagnostics, UK). The plasma samples were diluted 1:1 with water to ensure that all samples were below the maximum detection limit of the method (15 mM). All samples were analysed in duplicates.

**RNA extraction and qPCR.** Meninges, hippocampi, skeletal muscle (*triceps surae*), adipose tissue, liver and pancreas were quickly dissected out from wild-type and *Hcar1* knockout mice ($n = 5$ each, except for fat, liver and pancreas $n = 4$; meninges from five mice were pooled in one sample for wild-type and one for knockout) and snap-frozen in liquid nitrogen and stored at $-80\,°C$ until RNA extraction. The tissue was homogenized in Qiasol lysis buffer and lysing Matrix D tubes in a Fastprep machine, at speed 6.5 for 40 s (repeated for muscle tissue). RNA was extracted using the RNeasy Lipid Tissue Mini Kit (Qiagen), according to the manufacturer's protocol, and eluted in 30 μl RNase-free water. The Turbo DNAse (Ambion) kit was used for DNAse treatment of the extracted RNA, and the concentration and quality was measured using Nanodrop. Total RNA (2 μg) was used per cDNA reaction for all tissues except meninges (lower concentration, 1.1 μg total RNA for knockout and 1.3 μg total RNA for wild type) in a total volume of 20 μl, using the High Capacity RNA-to-cDNA Kit (Applied Biosystems by Thermo Fisher Scientific, 50 reactions kit). For qPCR, the cDNA was diluted to 5 ng μl$^{-1}$, and 10 ng used per well. Power SYBR Green PCR MasterMix (Applied Biosystems, 5 ml) was used according to the protocol, with TATA-box-binding protein as a reference gene. A StepOne machine was used to run the qPCR reactions using the 2 h standard programme. To verify primer specificity, the resulting qPCR products were run on a 2% agarose gel. In addition, to verify no genomic DNA contamination, melting curve analysis was performed and negative controls were included (water and without reverse transcriptase for each sample). TaqMan-based detection was used to examine the gene expression of HCAR1 in meninges relative to the expression in the hippocampus (Supplementary Table 1, experiment 2). An amount of 200 ng total RNA was used per cDNA reaction for hippocampus and meninges in a total volume of 10 μl, using Reverse Transcriptase Core kit RT-RTCK-05 (Eurogentec, Liège, Belgium). The cDNA was diluted to 5.7 ng μl$^{-1}$. Each qPCR reaction was prepared with Takyon Low ROX Probe Mastermix dTTP Blue UF-LPMT-B0701 (Eurogentec) according to the protocol, with 10 μl of diluted cDNA as template. An Agilent Mx3005P (Stratagene, La Jolla, CA, USA) machine was used to run the qPCR reactions. Ribosomal protein L27a (Rpl27a) was used as reference gene. The TaqMan probes used were Gpr81 Mm00558586_s1 and Rpl27a Mm00849851_s1, TaqMan Gene Expression Assays (Applied Biosystems by Thermo Fisher Scientific).

**Quantitative western blotting.** At 6 h after the end of the exercise, or 6 h after the last dose of L-lactate or saline, the mice were deeply anaesthetized with isoflurane and killed by decapitation. The hippocampi were carefully dissected out on ice and snap-frozen in liquid nitrogen. Material from *in vitro* experiments was processed similarly. For western blot analysis, lysates were prepared by homogenizing in radio-immunoprecipitation assay buffer (Sigma-Aldrich) containing cOmplete Protease Inhibitor Cocktail (Roche). Samples, 10 μg of total protein, were subjected to gel electrophoresis (12% SDS/PAGE (Bio-Rad)), transferred to a nitrocellulose membrane, using Transblot Turbo (Bio-Rad) and incubated overnight with the following primary antibodies: rabbit anti-VEGFA (ab46154; Abcam; diluted 1:1,000), mouse anti-α-tubulin (3873S; Cell Signaling Technologies (CST), Beverly, MA, USA; diluted 1:1,000), mouse anti-ERK1/2 (anti-p44/42 MAPK, 4696S; CST; diluted 1:1,000), rabbit anti-phospho-ERK1/2 (Thr202/Tyr204; anti-phospho-p44/42 MAPK, 4370; CST, diluted 1:1,000), mouse anti-Akt (pan; 4691P; CST; diluted 1:1,000), rabbit anti-phospho-Act (Ser473; 4060S; CST, diluted 1:1,000). Protein bands were visualized using horseradish peroxidase (HRP)-coupled species-specific secondary antibodies (GE Healthcare Life Sciences, Oslo, Norway), that is, Amersham enhanced chemiluminescence (ECL) anti-rabbit HRP-linked whole antibody from donkey (NA934; diluted 1:20,000) and Amersham ECL anti-mouse HRP-linked whole antibody from sheep (NA931; diluted 1:50,000), combined with a chemiluminescent detection system (SuperSignal West Dura Extended Duration, Thermo Scientific, Rockford, USA). After probing the blots with phospho-specific antibodies, the blots were stripped and reprobed with antibodies to total ERK1/2 and total Akt. All antibodies produced bands corresponding to the published molecular mass of the antigen, suggesting that the antibodies were specific. Quantification of the band density was performed using the Image Studio Lite

(Luke Miller) software, according to the published method (http://lukemiller.org/index.php/2013/02/analyzing-western-blots-with-image-studio-lite/).

**Immunohistochemistry and fluorescence microscopy.** At 6 h after the end of the exercise or 6 h after the last dose of lactate/saline, *Hcar1* knockout and wild-type mice were deeply anaesthetized with zolazepam 3.3 mg, tiletamine 3.3 mg, xylazine 0.5 mg, fentanyl 2.6 μg ml$^{-1}$; 0.1 ml 10 g$^{-1}$ bodyweight, intraperitoneally (i.p.) and transcardially perfused with 4% paraformaldehyde in 0.1 M sodium phosphate buffer pH 7.4 (NaPi) for 8 min. Non-treated *Hcar1* knockout and wild-type mice and mRFP-HCAR1 reporter mice were anaesthetized and perfused in a similar manner. Brains from some mRFP reporter mice were viewed without perfusion or further processing (Fig. 2a,b). After perfusion fixation, organs were gently removed, viewed with a stereomicroscope (for example, Fig. 2a,b), and stored in fixative at 4 °C until cutting 50-μm-thick vibratome sections or 20-μm-thick frozen sections (after cryoprotection by immersion in 30% sucrose in 0.1 M NaPi solution overnight). Free floating brain sections were washed three times in PBS(10 mM NaPi, 0.9% NaCl) and then subjected to fluorescence immunocytochemistry, directly, or after being incubated for 30 min in citrate buffer (0.01 M, pH 8.7) at 80 °C for antigen retrieval. The sections were rinsed in PBS and unspecific binding sites were blocked by incubating the sections with 10% newborn calf serum and 0.5% Triton X-100 in PBS for 4 h. The sections were then incubated with primary antibodies (shielded from light, on an orbital shaker, overnight): mRFP signal was enhanced by rat anti-mRFP (IgG2a monoclonal, ChromoTek GmbH, Germany, code 5F8; diluted 1:500), followed by Cy3 donkey anti-rat IgG (H + L) polyclonal affinity-purified (Jackson ImmunoResearch Laboratories Inc., USA; code 712-165-150; diluted 1:500). Endothelial cells were labelled using FITC rat anti-mouse CD31 (BD Biosciences; diluted 1:200). Vascular basement laminae were labelled by rabbit anti-collagen IV (Abcam; ab6586; diluted 1:500). Fibroblast-like cells were labelled with mouse anti-vimentin monoclonal IgG$_1$ (Santa Cruz Biotechnology; code E-5, sc-373717; 1:250). Pericyte-like cells were labelled with rabbit monoclonal anti-PDGFRβ (Abcam; ab32570, Y92; diluted 1:100). Antibodies raised in mouse or rabbit were followed by rinsing in PBS and incubation with Alexa Fluor 488 donkey anti-mouse IgG (H + L; code A21202; Molecular Probes/Thermo Fisher, Waltham, MA) or Alexa Fluor 488 donkey anti-rabbit IgG (H + L; code A21206), both diluted 1:400 overnight. In some experiments, DAPI (Molecular Probes, Eugene, OR) was used to counterstain cellular nuclei. The sections were rinsed and mounted with Prolong Gold Antifade reagent with DAPI (Life Technologies) and cover slipped (Assistant, Germany). For the triceps surae muscle, 14 μm transverse cryostat sections were cut and mounted on glass slides. The sections were labelled as described above, but double labelling of collagen IV and chicken anti-MCT1 (diluted 1:1,000) was performed. Secondary antibodies were Alexa Fluor 488 donkey anti-chicken IgG (H + L) and Alexa Fluor 555 donkey anti-rabbit IgG (H + L), both diluted 1:400.

Surface view images were taken by an epifluorescence dissecting microscope (SteREO Lumar.V12, Zeiss, Germany, or Leica MZ6, Leica, Germany). Images of sections were acquired using confocal laser scanning microscopes (Leica TCS SP5, Leica, or LSM 6 Pascal or LSM 510 Meta, Zeiss). For quantification of capillaries in the sensorimotor cortex (Fig. 1a,b) and cerebellar cortex (Supplementary Fig. 4) fluorescence images were acquired with an Axio Scan Z1 (Carl Zeiss), imaging whole-sagittal brain sections at a resolution of 0.11 μm per pixel (Norbrain Slidescanning Facility, http://www.med.uio.no/imb/english/research/about/infrastructure/norbrain/slidescanning/).

**Capillary density and diameter in the brain.** Parasagittal brain sections were labelled for collagen IV as described. For hippocampus, two confocal z-stacks, 5.48 μm thick, were obtained from two separate areas in each animal, covering the whole hilus. For sensorimotor cortex and cerebellar cortex, high-resolution fluorescence images of 20 μm sections were acquired, by means of an automated slide scanner system (Axio Scan Z1, Carl Zeiss Microscopy, Munich, Germany) in order to efficiently sample large areas of brain tissue. Images were inspected using the Zen Lite Blue software (Carl Zeiss Microscopy). The quantifications were performed by an observer who was blinded with regard to treatments and genotypes. Using the SimpleGrid plug-in for ImageJ, an array of points was overlaid on the image, the hilus area of the dentate gyrus was outlined (defined as bordering on the granule cells, an on a straight line connecting the two 'extremes' of the dentate granule cell layer), and the number of points over capillaries (within their outer borders) were counted and compared to the total number of points over hilus (points over larger vessels and artefacts subtracted) to calculate the fraction of area occupied by capillaries. Similarly, in the cerebral cortex, the whole-cortical thickness between pia mater and white matter was sampled from 2 mm anterior to 2 mm posterior of bregma, and in the cerebellum, all three cortical layers from the entire folia 1–2 in lobus anterior were sampled. According to the Delesse Principle, the ratio of the number of points that hit the sectional profiles of the object (capillaries), to the total number of points probed on the sections through a struucture equals the ratio of the volume of the object to the volume of the entire structure[49]. The average capillary diameter and the section thickness affect the absolute values obtained, but not the relative changes between experiment and control (see Supplementary Table 2). Capillaries to be included in the analysis were defined as vessels no more than 10 μm in diameter[24]. In each mouse, the external diameter was measured in at least 10 capillaries with a visible

lumen, using the ImageJ software, and averaged. The data are presented as mean ± s.e.m. of four to seven mice per group, as specified.

**Capillary density and fibre diameter in muscle.** Transverse sections of *triceps surae* muscle, labelled for collagen IV (capillary) and MCT1 (muscle fibre) as described above, were examined for total area of capillaries per total area of muscle fibres, and for area of individual muscle fibres (about 30 in each condition) using ImageJ. The data are presented as mean ± s.e.m. of five to six mice in each group.

**Fluorescent dye angiography.** To visualize brain vessels (Fig. 2e), jugular veins were retrogradely infused *in situ* following fixation using 1% low melting agarose (Sigma) with 50 mg ml$^{-1}$ fluorescein-labelled 500 kDa dextran (Molecular Probes).

**Two-photon imaging.** In order to perform two-photon measurements *in vivo*, a cranial window was implanted over the barrel field of the somatosensory cortex in two of the mRFP-HCAR1-expressing mice, both of which showed the described cellular expression. During imaging, the mice were head-fixed and freely walking on a spherical treadmill[50]. We imaged the mRFP-HCAR1-expressing cells in the pia mater using a two-photon laser scanning microscope (Prairie Ultima IV, Bruker Corporation, USA). The fluorophore was excited by a Ti:sapphire laser (InSight, Spectra-Physics, USA), which was set to emit laser pulses at a wavelength of 1,000 nm, and the laser pulses were focused on the sample through a water immersion objective (N16XLWD-PF, numerical aperture (NA) 0.8, Nikon, Japan). The fluorescent signal was then filtered through a bandpass filter with a 50 nm bandwidth ∼525 nm (ET525/50 m, Chroma, USA) and captured in a photomultiplier tube. Images were acquired in the PrairieView software (Bruker Corporation).

**HCAR1 immunocytochemistry.** Mice (wild-type, *Hcar1* knockout and mRFP-HCAR1 reporter) were anaesthetized and transcardially perfused with 4% paraformaldehyde in PBS before brain removal and post fixation in the same fixative for 24 h (ref. 51). Following dehydration in ethanol and embedding in paraffin, 4 µm thick sagittal sections through the entire forebrain were cut with a microtome (Thermo Scientific). Test sections were deparaffinized in Neoclear (Millipore) followed by rehydration through an ethanol gradient (100%, 3 min; 100%, 3 min; 96%; 1 min and 70% 1 min) and subsequently incubated at 100 °C, under pressure of $0.8^{-1}$ bar in citrate antigen retrieval buffer (pH 6.0), containing 0.05% Tween 20 (Sigma) for 2 min. Test sections were then washed in Milli-Q water and PBS containing 0.1% Tween 20. Additional sagittal brain sections derived from *Hcar1* knockout mice were prepared as described above and used to pre-absorb the primary antibody solution, containing anti-HCAR1 antibody (SAB1300090 SIGMA anti-mouse Gpr81-s296 antibody produced in rabbit, affinity-isolated; 1:200 dilution), 10% goat serum, 10% BSA and 0.1% Tween 20 for 1 h at room temperature. The pre-absorbed primary antibody solution was subsequently transferred on test wild-type and *Hcar1* knockout sections, mounted on microscopic slides (Thermo scientific), and incubated overnight in a humid atmosphere at 4 °C. After three washes on shaker in PBS containing 0.1% Tween 20 (5 min each), staining with Alexa Fluor 555 secondary antibody (Invitrogen) was performed in a solution containing 0.5% BSA, 0.5% goat serum and 0.1% Tween 20 for 90 min at 37 °C. After three washes in 0.1% Tween 20 in PBS (5 min each) on a shaker, samples were mounted in Wectashield mounting medium containing DAPI (Vector Laboratories). Microscopy was carried out using a Leica SP8 three-dimensional confocal microscope equipped with a ×20 water immersion lens (NA 0.75).

**Hippocampal slices.** Hippocampal slices were prepared from *Hcar1* knockout and wild-type mice and incubated in Krebs buffer as described before[18], with additions of L-lactate or the selective HCAR1 agonist 3,5-dihydroxybenzoate[52], applied from neutral solution at the concentrations and for the times indicated (Supplementary Fig. 11). After incubation, the slices were extracted and processed for quantitative immunoblotting as described above.

**Statistics.** All data are shown as mean ± s.e.m. or ± s.d. as indicated. One-way analysis of variance followed by Fisher's least significant difference *post hoc* test was performed for comparison of the confocal data, Dunnett's T3 test when variance was unequal (in muscle). For statistical analysis of western blot data, where the genotypes were analysed separately, Dunnett's T3 *post hoc* test was used. $P < 0.05$ was considered statistically significant.

**Data availability.** The data sets generated during and/or analysed during the current study are available from the corresponding authors on reasonable request.

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

## Acknowledgements

This work was supported by grants from the Research Council of Norway (L.H.B., J.S.-M., K.A.A., C.M. and J.E.R.), the MLS^uio (C.M.), the Faculty of Dentistry, University of Oslo (Ø.P.H.), Lundbeckfonden, Denmark (E.H.D.) and Demensforbundet, Nasjonalforeningen, Norway (J.E.R.). We thank Sumaya Mohamed Haji Yusuf for help with obtaining the data presented in Supplementary Fig. 2, Erlend A. Nagelhus for the gift of antibody to PDGFRβ and Rajikala Suganthan for paraffin embedding. Images for capillary analysis of cerebral cortex and cerebellum were acquired at the Norbrain Slidescanning Facility at the Institute of Basic Medical Sciences, University of Oslo, a resource funded by the Research Council of Norway; this help is gratefully acknowledged.

## Author contributions

C.M., K.A.A., Ø.P.H., O.M. and A.H. performed the *in vivo* studies. C.M. and K.A.A. planned these studies and harvested organs. K.A.A., V.P., Y.C., A.H. and L.H.K. performed immunohistochemistry of the brain. K.A.A. and C.M. performed western blotting analysis and quantified the density of blood vessels in the brain. Ø.P.H., L.K. and E.H.D. performed mRNA analysis. J.E.R. performed immunolabelling and analysis of vessels in skeletal muscle. L.K., V.P. and M.P. tested antibodies against HCAR1. S.O. developed the knockout mouse; A.G. and S.O. developed the transgenic reporter mouse; and A.G., A.H., K.A.A. and L.K. performed confocal imaging of the reporter mice. T.S. and U.W. developed and validated the exercise regime. C.M. performed plasma lactate measurements. E.H. and K.V. performed two-photon microscopy. L.H.B., J.S.M., M.B. and C.M. designed the study. C.M. wrote the draft manuscript. All authors discussed the results and critically revised the final version of the manuscript. C.M. and K.A.A. contributed equally to the paper.

## Additional information

**Competing interests:** The authors declare no competing financial interests.

