## [Peer Review File · Nature Communications]

Reviewers' comments:

Reviewer #1 (Remarks to the Author):

Exercise induces cerebral angiogenesis via lactate receptor at pial vessels
By Cecilie Morland, ..., Linda Bergersen

This study investigates the possible role of lactate on angiogenesis in brain. The main claim of the article is that lactate binding on the receptor hydroxycarboxylic acid receptor 1 stimulates vascular endothelial growth factor A which in turn activates angiogenesis.

There is quite a body of evidence that lactate has an angiogenic effect in peripheral wound healing and the topic is of general interest and the study is original. However, the study lacks in part more rigorous methodology and more data on some of the subparts are needed. The paper is easy to read and appears clear.

1. The studies cited in the beginning of the introduction make a link between exercise and cognitive performance, particularly in the elderly population. However, the link to lactate is not at all apparent. The authors must more clearly make this explicit. There are many other possible pathways – completely independent from lactate and its receptor) - involved in these effects.

2. The authors provide vascular volume fractions (VV) (e.g. Figure 1) in percent of total volume. These values are too high compared to the literature. It is completely unclear how the authors computed vascular volume fraction from their microscope images of the anti-collagen stainings. A VV quantification is possible, but not straight forward. In addition to vascular volume fraction, it would be interesting to know the length density and the calibers of the vessels. A stereological approach as explained in *Cereb Cortex*. 2008 18(10):2318-30 would yield these data. However, the analysis requires a very good quality of the anti-collagen staining.

3. The volume fraction (Figure 1) rises from 8% (which is already high, see above) to 12% which is really too high and this increase is extraordinary and very hard to believe.

4. Blood plasma lactate levels are crucial for this study. The supplemental figure 1 shows the plasma lactate kinetics after the subcutaneous shots of lactate versus PBS. However, the lactate levels are way too high. The PBS treated mice show a lactate level of higher than 5 after 180 minutes! This is about 5-times too high. Something is clearly wrong here, as plasma levels of around 1 mM are expected. These data, when re-visited, belong in the main part of the paper.

5. The authors make use of in vivo two-photon microscopy to show the localization of mRFP-HCAR1. It would be really important to corroborate the vascular density data (see my point 3) with in vivo two-photon stacks of a fluorescent dextran conjugate filled vessels. I guess these data are available anyways and would make a strong case in confirming the histological vascular density analysis, which in my view is not yet done well (see my point 2).

Reviewer #2 (Remarks to the Author):

This is quite an interesting manuscript that describes the pial vessel distribution of the lactate receptor HCAR1 (or GPR81) and impact of lactate stimulation on angiogenesis within the brain. They have developed powerful tools to see the distribution and impact of HCAR1. The use of a HCAR1 fluorescent reporter protein allowed them to show that the leptomeningeal cells of the pia mater express this receptor. Also the development of the HCAR1 knockout transgenic mouse allowed them to show that

lactate stimulation leads to angiogenesis selectively in the brain. Most remarkably they also could show that the exercise dependent angiogenesis in the brain requires the HCAR1 receptor and is mimicked by lactate itself. This work is convincing and well executed. It is a beautiful study that provides a remarkable insight into the possible mechanisms of the improvement in brain health from exercise. I agree with the authors that the work points to the importance of developing pharmacological agonists to this receptor.

Minor point: The impact of rosiglitazone is intriguing and leads to increased expression of HCAR1. However the distribution of HCAR1 looks like it might be slightly different. The example shown in Fig 3 i-j shows staining in cells surrounding capillaries in superficial layers of the cortex. The cell looks remarkably like a pericyte and in some of the other images it is difficult to determine whether a subpopulation of pericytes might be stained. Do the authors have any data on expression in pericytes? This could easily be checked using PDGFR-beta and/or NG2 antibodies.

I am a little puzzled by the experiment in the hippocampal slices. Although I am aware of the previous work by this group showing some cellular expression of HCAR1 in the hippocampus this study only shows some expression in blood vessels in the hippocampal fissure. Are the authors proposing that the signaling pathways triggered by lactate are due to the vessel responses or are they detecting effects of stimulating the receptors expressed at low levels in the cells within the hippocampus. The authors should also describe a bit more about what they are proposing. Are the leptomeningeal cells detecting lactate and releasing VEGFA to parenchyma to stimulate angiogenesis in regions downstream from the pial vessels. This is implied by the diagram but should be explicitly stated.

Reviewer #3 (Remarks to the Author):

In the paper from Morland C et al., the authors hypothesize that lactate released after intense exercising induces cerebral angiogenesis via the lactate receptor (HCAR1) at pial vessels. The hypothesis is very interesting and the authors present some compelling preliminary data (that knockout of HCAR1 suppresses exercise-induced cerebral angiogenesis), however the data presented are incomplete such that it is not possible to draw the suggested conclusions.

- While the knockout does demonstrate that HCAR1 is the key receptor, there is no evidence that its action is required in the pial vessels. The authors show that HCAR1 is expressed in the cells associated with the pial vessels, but also say that it is expressed in other cells including neurons. It is also very possible that these other cells are the cell type that is responsible for mediating the lactate-HCAR1 interaction. Therefore without direct evidence that the pial vessels cells are the cells responsible it is impossible to make the conclusion that is in that title of the paper. Furthermore, there is evidence to suggest that it may not be the pial vessels. For instance, they demonstrate changes in the angiogenesis in the hippocampus which is a brain region at a distance from the vessels. It is unclear how the signal from the pial vessels would be propagated to the hippocampus. Furthermore, they perform experiments on the hippocampus in slice cultures, yet it is not clear whether there are even pial vessels present in these hippocampal slice cultures. To claim that the interaction between lactate and angiogenesis is mediated through pial vessels the authors need to perform cell specific knockouts or direct cell biochemistry to identify the key site of cellular action.

- In experiments 1E,F the authors aim to demonstrate that lactate and exercise increase hippocampal VEGFA and that this is suppressed in the HCAR1 knockout. If one looks at the data though it is clear that the baseline VEGFA is higher in the HCAR1 knockouts compared to controls. When there is an exercise/lactate-dependent increase in the VEGFA in the wild type, it only reaches the baseline levels of the HCAR1 mice. Therefore, the lack of exercise/lactate-dependent increase in the HCAR1 KO mice

may derive from the fact that the baseline is already high and potentially saturated as opposed to an important role of HCAR1 in regulating and exercise/lactate response. This is a really interesting observation but the authors need to evaluate what this means in terms of their experimental paradigm. Furthermore, in figure 1b it appears that there is higher baseline vascular density in the motor cortex of the HCAR1 knockout mice compared to controls, once again making it difficult to determine how exercise and lactate may act through this receptor.

- The authors show an exercise/lactate-dependent increase in cerebral angiogenesis in the M1 cortex and hippocampus and an exercise/lactate dependent increase VEGFA in the hippocampus. This leaves many questions. Is there not an increase in the VEGFA in the motor cortex? What other brain regions show an increase in cerebral angiogenesis. The authors need to survey several brain regions to determine if this is a region-specific response or a pan-brain response.

- The authors need to show where HCAR1 is expressed by doing the appropriate staining. The tested antibodies listed work in paraffin-embedded tissue. The authors should stain by immunohistochemistry instead of immunofluorescence.

- The rosiglitazone experiment is not particularly useful. The authors use this drug to elevate HCAR1 expression to determine where it is expressed. The drug, however, may cause a change in the expression of the receptor as opposed to just an increase in the cells that already express it.

Reviewer #4 (Remarks to the Author):

This is an interesting paper that purports to show that the activation of lactate receptors at pial vessels results in cerebral angiogenesis. The authors provide data from diverse protocols to support their conclusions. In general the paper is well written and the data presented clearly. The claim that increased circulating lactate can induce angiogenesis is somewhat novel and certainly intriguing with far reaching therapeutic consequences if validated. There are, however, a number of critical issues that should be addressed.

1. The most important issue concerns the methodology for determining angiogenesis, which underlies the major point of the paper. This methodology is not discussed in any detail. It appears to be based on a ratio of fluorescence positive (to Collagen IV antibody) pixels to tissue background pixels and presented as a volume percent. There are several concerns. A) Does this method distinguish changes in dilation? B) Does the method account for regional and cortical layer differences in capillary density? C) How were sampling errors minimized in such heterogeneous tissues? D) Were the data recorded by blinded investigators? E) The volume percent reported (approx.. 8%) is about double that expected for average cerebral cortex (the hippocampal data are more in line with expectations). F) It is not clear which vessels were included in the measurement (arterioles?, capillaries?, venules?).

2. The investigators follow up their initial findings with experiments designed to test potential molecular mechanisms. But, it is unclear how activation of pial associated cells would directly affect capillary endothelial cells at a distance. Even though changes may be detected, it is not possible to conclude that these were the only changes, or that a cause and effect can be shown.

3. There are no error bars provided in Supplemental figure 2.

4. The data shown in supplemental figure 3 do not show that the responses of the different groups were the same, only that no statistical differences were detected. It is likely that the test does not have the power to discriminate any differences. The authors gloss over this by using a larger than necessary y-scale scale and error bars based on s.d., whereas all other graphs use s.e.m.

5. Claiming statistical significance to p equals 0.01 or smaller is not justified by the numbers of observations because reasonable test power cannot be maintained. (Button, K. S., et al. (2013).

"Power failure: why small sample size undermines the reliability of neuroscience." Nat Rev Neurosci

14(5): 365-376.) $P < 0.05$ should be sufficient to make the point.

6. In the analysis of the data from supplementary figure 8, how did the investigators control for changes in fiber volume, which is a well known mechanism for decreasing intercapillary distance and increase capillary density without angiogenesis?

RESPONSE TO REFEREES

Reviewers' comments:

Reviewer #1 (Remarks to the Author):

Exercise induces cerebral angiogenesis via lactate receptor at pial vessels

By Cecilie Morland, ..., Linda Bergersen

«This study investigates the possible role of lactate on angiogenesis in brain. The main claim of the article is that lactate binding on the receptor hydroxycarboxylic acid receptor 1 stimulates vascular endothelial growth factor A which in turn activates angiogenesis. There is quite a body of evidence that lactate has an angiogenetic effect in peripheral wound healing and the topic is of general interest and the study is original. However, the study lacks in part more rigorous methodology and more data on some of the subparts are needed. The paper is easy to read and appears clear.»

RESPONSE: *We thank the Reviewer for the positive comments and now add information on more rigorous methodology and more data, as shown below.*

«1. The studies cited in the beginning of the introduction make a link between exercise and cognitive performance, particularly in the elderly population. However, the link to lactate is not at all apparent. The authors must more clearly make this explicit. There are many other possible pathways – completely independent from lactate and its receptor) - involved in these effects.»

RESPONSE: *The Reviewer is right that there are many possible pathways independent of lactate and its receptor, and we actually refer to a number of those in the second paragraph of the introduction. We agree with the Reviewer that the link to lactate should be strengthened by referring to its angiogenic effects in wounds and now add references on lactate in peripheral wound healing (Porporato PE et al 2012 Angiogenesis; Ruan GX, Kazlauskas A 2013 J Biol Chem).*

«2. The authors provide vascular volume fractions (VV) (e.g. Figure 1) in percent of total volume. These values are too high compared to the literature.»

RESPONSE: *The percentage of volume occupied by vessels given in the previous version of the manuscript might represent an overestimate, since we recorded the percentage of points over vessels compared to the number of points over the total area observed in a projection of a z-stack, without correcting for the thickness of the stack. (According to the Delesse Principle, the volume fraction equals the area fraction, assuming zero section thickness.) Further, the external limit of staining for a basal lamina marker (collagen IV), rather than the capillary lumen was recorded. (The external limit was chosen because it gave a more precise definition than the lumen.)*

According to the reference cited below by the Reviewer #1 (Weber B et al 2008 Cereb Cortex), in its Appendix (p 2328), the volume fraction of capillaries (V_F) can be calculated from their mean diameter (d_V), their area fraction (A_F) and the section thickness (t) as $V_F = d_V \cdot A_F / t$. We have now determined d_V to be about $5.8 \mu\text{m}$ (outer diameter, legend of new Fig.1), while A_F is about 7%, in the hippocampal hilus in wild-type controls (previous

Fig. 1d), and t equals the z-stack thickness, 5.48 μm . This gives $V_F = 5.8 \mu\text{m} * 7\% / 5.48 \mu\text{m} = 7.4\%$, i.e., correction to a slightly higher rather than a lower value.

We are not aware which specific publications in the literature the Reviewer has in mind to support the statement that our values for vascular volume fraction are «too high». However, our value of 7% agrees reasonably well with literature data, which vary quite a bit among different studies: Zerbi V et al 2013 Brain Struct Funct, who estimated the capillary density from the fraction of area covered by staining of capillary endothelial glucose transporter-1, reported a value of 12% in the dentate gyrus of wild-type mice. Heinzer S et al 2008 NeuroImage, who performed 3D analysis of intravascular casts after etching away the tissue, reported 3% capillary volume fraction in hippocampus of wild-type mice (their Fig. 3a). Desjardins M et al 2014 Neurobiol Aging, who used two-photon microscopy of intravascular fluorescent dextran in live young rats, estimated the capillary volume fraction in the sensorimotor cortex at $6.8 \pm 0.3 \%$.

Importantly, the purpose of the current study is not to determine the absolute capillary density in the brain, but rather to investigate the effect of the treatments on vascularization, for which our assessment is valid. It was done in exactly the same way in all of the mice. Thus correction for section thickness would cancel out when values are calculated relative to controls. To avoid confusion, we have now normalized our data relative to the wild-type control group.

«It is completely unclear how the authors computed vascular volume fraction from their microscope images of the anti-collagen stainings.»

RESPONSE: We apologize for this lack of clarity. We have now explained the analysis in the Methods description as follows: “**Capillary density and diameter in brain.** Parasagittal brain sections were labelled for collagen IV as described. Two z-stacks 5.48 μm thick were obtained from two separate areas in each animal, covering the whole hilus area in hippocampus. For sensorimotor cortex and cerebellar cortex, high-resolution images of histological sections were acquired using an automated slide scanner system (Axio Scan Z1, Carl Zeiss Microscopy, Munich, Germany). Images were inspected using the Zen Lite Blue software (Carl Zeiss Microscopy). The quantifications were performed by an observer who was blinded with regard to treatments and genotypes. Using the SimpleGrid plug-in for ImageJ, an array of points was overlaid on the image, the hilus area of the dentate gyrus was outlined (drawing a straight line connecting the two ‘extremes’ of the dentate granule cell layer), and the number of points over capillaries (within their outer borders) were counted and compared to the total number of points over hilus (points over larger vessels subtracted) to calculate the fraction of area occupied by capillaries. Similarly, in the cerebral cortex, the whole cortical thickness between pia mater and white matter were sampled from 2 mm anterior to 2 mm posterior of bregma, and in cerebellum, all three cortical layers from the entire folia 1-2 in lobus anterior were sampled. According to the Delesse Principle, the fraction of area occupied by a structure equals the fraction of the volume (e.g., West MJ 2012 Cold Spring Harb Protoc doi: 10.1101/pdb.top070623). Capillaries to be included in the analysis were defined as vessels no more than 10 μm in diameter (Hall CN et al 2014 Nature). In each mouse, the external diameter was measured in at least 10 capillaries with a visible lumen, using the ImageJ software, and averaged. The data are presented as mean \pm s.e.m. of 4-7 mice per group, as specified.”

«A VV quantification is possible, but not straight forward. In addition to vascular volume fraction, it would be interesting to know the length density and the calibers of the vessels.»

RESPONSE: *We have now measured the diameter (calibre) of the capillaries, and found that it did not differ between the groups. This indicates that the increase in vascular density observed in hippocampus represents an increase in the area of contact between blood and brain parenchyma. These data are given in the legend of the revised Fig. 1, and stated in the revised text.*

«A stereological approach as explained in Cereb Cortex. 2008 18(10):2318-30 would yield these data. However, the analysis requires a very good quality of the anti-collagen staining.»

RESPONSE: *We thank the Reviewer #1 for giving this reference, which we have used in our response to one of the previous comments. We have reexamined the preparations for capillary calibre (see above). The results on calibre are presented in the legend of the revised Fig. 1 and in the new Supplementary Fig. 4c.*

«3. The volume fraction (Figure 1) rises from 8% (which is already high, see above) to 12% which is really too high and this increase is extraordinary and very hard to believe.»

RESPONSE: *As described above, we agree that the absolute numbers given in the previous version of the manuscript may be overestimated, although similar values appear in the literature. However, the relative values are valid for comparison among the experimental groups. Indeed, the ca 40% increase in capillary area fraction reported here, is in line with the 49% increase (in 'vascular surface area', corresponding to the area fraction recorded here) that was reported by van Praag and coworkers after exercise (van Praag H, Shubert T, Zhao C & Gage FH 2005 J Neurosci; their Fig. 3b).*

«4. Blood plasma lactate levels are crucial for this study. The supplemental figure 1 shows the plasma lactate kinetics after the subcutaneous shots of lactate versus PBS. However, the lactate levels are way too high. The PBS treated mice show a lactate level of higher than 5 after 180 minutes! This is about 5-times too high. Something is clearly wrong here, as plasma levels of around 1 mM are expected. These data, when re-visited, belong in the main part of the paper.»

RESPONSE: *We are not aware of which published articles the Reviewer uses as basis for suggesting that our values are too high and that a value of 1 mM of plasma lactate is expected. Perhaps the Reviewer has in mind fasting levels of lactate? These are lower than fed values. Thus Wang MW et al 1992 Endocr Res found basal lactate values of 2.16 ± 0.18 mM and 3.10 ± 0.29 mM in fasted and fed mice, respectively. It should be noted that our mice were not fasted. Further, Iversen NK et al 2012 Respir Physiol Neurobiol ('The normal acid-base status of mice') took small blood samples from undisturbed mice with indwelling catheters in the carotid artery, finding lactate concentrations of 4.6 ± 0.7 mM. This is in excellent agreement with the values in our Supplementary Fig. 2. Similarly, Suhara T et al 2015 PNAS reported 3.7 mM (their Fig. 1B), and Flynn JM et al 2010 PLoS One reported 3.2 mM in wild-type control mice. Our control lactate value of 3.05 ± 0.72 mM (Supplementary Fig. 2) is therefore fully in agreement with literature data.*

We still think the figure in question belongs in the Supplementary material, because it presents background information rather than novel results, but leave the decision on this matter to the Editor's discretion.

«5. The authors make use of in vivo two-photon microscopy to show the localization of mRFP-HCAR1. It would be really important to corroborate the vascular density data (see my point 3) with in vivo two-photon stacks of a fluorescent dextran conjugate filled vessels. I guess these data are available anyways and would make a strong case in confirming the histological vascular density analysis, which in my view is not yet done well (see my point 2).»

RESPONSE: *The trained and lactate injected mice were not examined by two-photon microscopy, and it does not seem reasonable to repeat this experiment with mice that have cranial windows for two-photon examination. As explained above, it is the relative change in vascular volume that is important for the conclusions of the paper, not the absolute values.*

However, there are data in the literature on capillary volume examined by two-photon microscopy in vivo. Above we already quoted the paper by Desjardins M et al 2014 Neurobiol Aging, who used two-photon microscopy of intravascular fluorescent dextran in live young rats, and estimated the capillary volume fraction in the sensorimotor cortex at 6.8 ± 0.3 %, and the internal capillary diameter at 6.2 ± 0.1 μm . These values are in agreement with our data in mouse hippocampus (7% and 6 μm , respectively).

We do hope the Reviewer will agree with us on our responses to the issues raised.

Reviewer #2 (Remarks to the Author):

«This is quite an interesting manuscript that describes the pial vessel distribution of the lactate receptor HCAR1 (or GPR81) and impact of lactate stimulation on angiogenesis within the brain. They have developed powerful tools to see the distribution and impact of HCAR1. The use of a HCAR1 fluorescent reporter protein allowed them to show that the leptomeningeal cells of the pia mater express this receptor. Also the development of the HCAR1 knockout transgenic mouse allowed them to show that lactate stimulation leads to angiogenesis selectively in the brain. Most remarkably they also could show that the exercise dependent angiogenesis in the brain requires the HCAR1 receptor and is mimicked by lactate itself. This work is convincing and well executed. It is a beautiful study that provides a remarkable insight into the possible mechanisms of the improvement in brain health from exercise. I agree with the authors that the work points to the importance of developing pharmacological agonists to this receptor.»

RESPONSE: *We appreciate that the Reviewer finds our work to be convincing and well executed, beautiful, and providing remarkable insight, and are grateful for the positive comments on the pharmacological potential.*

«Minor point: The impact of rosiglitazone is intriguing and leads to increased expression of HCAR1. However the distribution of HCAR1 looks like it might be slightly different. The example shown in Fig 3 i-j shows staining in cells surrounding capillaries in superficial layers of the cortex. The cell looks remarkably like a pericyte and in some of the other images it is difficult to determine whether a subpopulation of pericytes might be stained. Do the authors have any data on expression in pericytes? This could easily be checked using PDGFR-beta and/or NG2 antibodies.»

RESPONSE: *In view of the remarks by Reviewers #2 and #3, we have now deleted the rosiglitazone data. New results with PDGFR- β are now presented in the new Fig. 3j,k,l,*

which shows colocalization with mRFP-HCAR1, thus suggesting that HCAR1 is expressed in (a subpopulation of) pericytes. This is now described and commented on in the text.

«I am a little puzzled by the experiment in the hippocampal slices. Although I am aware of the previous work by this group showing some cellular expression of HCAR1 in the hippocampus this study only shows some expression in blood vessels in the hippocampal fissure. Are the authors proposing that the signaling pathways triggered by lactate are due to the vessel responses or are they detecting effects of stimulating the receptors expressed at low levels in the cells within the hippocampus. The authors should also describe a bit more about what they are proposing. Are the leptomeningeal cells detecting lactate and releasing VEGFA to parenchyma to stimulate angiogenesis in regions downstream from the pial vessels. This is implied by the diagram but should be explicitly stated.»

RESPONSE: In the legend of the schematic Fig. 5 (previously Fig. 4), we now describe the possibilities mentioned by the Reviewer #2: Magnified inset indicates possible, yet unidentified pathways leading from activation of HCAR1 in the leptomeningeal cells in pia mater and perivascular sheaths to increased VEGFA and subsequent enhanced angiogenesis. Although not clearly labelled for mRFP-HCAR1, other cells may possibly express low levels of HCAR1. HCAR1 may stimulate VEGFA in the same cells, or in other cells, through mediators. Importantly, blood to the brain parenchyma has to pass close by the perivascular sheath of HCAR1 carrying cells, which covers all blood vessels to the brain, and therefore may convey products released from these cells upon activation of the receptor; blood to the hippocampus passes through vessels entering in the hippocampal fissure, such as the ones shown in the new Fig. 3i.

Reviewer #3 (Remarks to the Author):

«In the paper from Morland C et al., the authors hypothesize that lactate released after intense exercising induces cerebral angiogenesis via the lactate receptor (HCAR1) at pial vessels. The hypothesis is very interesting and the authors present some compelling preliminary data (that knockout of HCAR1 suppresses exercise-induced cerebral angiogenesis), however the data presented are incomplete such that it is not possible to draw the suggested conclusions.»

RESPONSE: We appreciate that the Reviewer #3 finds our hypothesis very interesting. The shortcomings mentioned by the Reviewer have now been amended to make the conclusions fully valid.

«- While the knockout does demonstrate that HCAR1 is the key receptor, there is no evidence that its action is required in the pial vessels.»

RESPONSE: New pictures are now shown, underlining the fact that HCAR1 carrying blood vessels extend from the pia all the way into the hippocampus, where the effects on VEGFA and angiogenesis are demonstrated. However, we agree that the action might also take place through HCAR1 located elsewhere than on these vessels and have therefore deleted the term "at pial vessels" from the title.

«The authors show that HCAR1 is expressed in the cells associated with the pial vessels, but also say that it is expressed in other cells including neurons. It is also very possible that these other cells are the cell type that is responsible for mediating the lactate-HCAR1 interaction.»

RESPONSE: *We do not actually say that the HCARI is expressed in other cells than those associated with vessels. But as the mRFP-HCARI expression cannot be expected to reveal all sites, the presence of HCARI at other sites cannot be excluded. We have therefore deleted "at pial vessels" from the title and have mentioned possible other sites in the description of the schematic in the new Fig. 5 (previous Fig.4).*

«Therefore without direct evidence that the pial vessels cells are the cells responsible it is impossible to make the conclusion that is in that title of the paper. Furthermore, there is evidence to suggest that it may not be the pial vessels. For instance, they demonstrate changes in the angiogenesis in the hippocampus which is a brain region at a distance from the vessels. It is unclear how the signal from the pial vessels would be propagated to the hippocampus.»

RESPONSE: *We have now amended these shortcomings, as stated in the two preceding RESPONSEs, deleting "at pial vessels" from the title. Of note, we show a new figure (Fig. 3i) demonstrating more clearly that the HCARI carrying vessels penetrate from the pia right into the hippocampus. It is not accurate to describe the hippocampus as "a brain region at a distance from the vessels"; in fact, the hippocampus is intimately enclosed by the pia mater, which penetrates deep into the hippocampal fissure between the cornu ammonis (CA1-CA3) and the gyrus dentatus, carrying the blood vessels that supply the whole hippocampal formation. Moreover, as also explained above, all blood to the brain parenchyma arrives via vessels running in the pia and therefore has to pass in close proximity to the HCARI carrying cells.*

«Furthermore, they perform experiments on the hippocampus in slice cultures, yet it is not clear whether there are even pial vessels present in these hippocampal slice cultures. To claim that the interaction between lactate and angiogenesis is mediated through pial vessels the authors need to perform cell specific knockouts or direct cell biochemistry to identify the key site of cellular action.»

RESPONSE: *Our experiments were not carried out in slice cultures, but rather in acute hippocampal slices, on which the pia is attached. Furthermore, HCARI carrying vessels, such as the ones now shown within the hippocampus in Fig. 3i, will be present also in the hippocampal slices incubated in vitro. But, again as already stated, we cannot exclude the possibility that HCARI sites on other cells contribute. This point has been taken care of by the amendments made (see preceding three RESPONSEs). Performing cell specific knockouts or direct cell biochemistry does not seem to be a reasonable requirement and would not change our key conclusion, namely that HCARI mediates exercise induced changes in VEGFA and angiogenesis.*

«In experiments 1E,F the authors aim to demonstrate that lactate and exercise increase hippocampal VEGFA and that this is suppressed in the HCARI knockout. If one looks at the data though it is clear that the baseline VEGFA is higher in the HCARI knockouts compared to controls. When there is an exercise/lactate-dependent increase in the VEGFA in the wild type, it only reaches the baseline levels of the HCARI mice. Therefore, the lack of exercise/lactate-dependent increase in the HCARI KO mice may derive from the fact that the baseline is already high and potentially saturated as opposed to an important role of HCARI in regulating and exercise/lactate response. This is a really interesting observation but the authors need to evaluate what this means in terms of their experimental paradigm.»

RESPONSE: The quantifications shown in the previous Fig. 1e are not comparable to the ones in the previous Fig. 1f, since these were run on separate Western gels. That is the reason why they are given as two figures instead of in one figure. The apparent differences between knockout and wild-type at baseline, represent differences in the saturation of the blot for the loading control (α -tubulin). The internal differences measured in each experiment, however, can be compared. The selective effects and increase over baseline are also illustrated by the pictures of the Western blots (Fig. 1g). To avoid confusion, we now present data that are normalized to the respective control groups (present Fig. 1e and Fig. 1f).

«Furthermore, in figure 1b it appears that there is higher baseline vascular density in the motor cortex of the HCAR1 knockout mice compared to controls, once again making it difficult to determine how exercise and lactate may act through this receptor.»

RESPONSE: We now present new analyses (revised Fig. 1a,b), of a much larger area, i.e., all layers of the whole sensorimotor cortex extending 4 mm anteroposteriorly in sagittal sections (rather than as before restricted to the 3 superficial layers of the motor cortex M1). The new data show highly robust exercise and lactate induced increases of capillary density in the wild-type, but not in the knockout mice.

«- The authors show an exercise/lactate-dependent increase in cerebral angiogenesis in the M1 cortex and hippocampus and an exercise/lactate dependent increase VEGFA in the hippocampus. This leaves many questions. Is there not an increase in the VEGFA in the motor cortex?»

RESPONSE: As stated in the preceding RESPONSE, we made new analyses, which are now presented in the revised Fig. 1a,b. We do not have material from cortex that can be analysed for VEGFA. (To do this analysis, would imply repeating all of the work, which does not seem reasonable.) However, we now present capillary density and VEGFA data from cerebellum (see below).

«What other brain regions show an increase in cerebral angiogenesis. The authors need to survey several brain regions to determine if this is a region-specific response or a pan-brain response.»

RESPONSE: Our key observation is increased VEGFA and angiogenesis in the hippocampal dentate hilus and in the sensorimotor cortex. The sampled area comprises a large proportion of the neocortex. While it would be interesting to know whether other brain regions show angiogenesis, we do not think that this is required for concluding that the hippocampal and neocortical changes are real. However, the answer is “no” to the Reviewer’s question of whether this is a pan-brain response: we have now done analyses of the cerebellum, which show no changes in VEGFA or angiogenesis (Supplementary Figs. 4 and 5).

«- The authors need to show where HCAR1 is expressed by doing the appropriate staining. The tested antibodies listed work in paraffin-embedded tissue. The authors should stain by immunohistochemistry instead of immunofluorescence.»

RESPONSE: We have now tested a commercial antibody (the same company and designation as that used in our previous publication, Lauritzen KH et al 2014 Cereb Cortex, on paraffin sections. In addition to this suggestion by the Reviewer #3, we preabsorbed the antibody with brain sections from HCAR1 knockout mice in order to remove immunoglobulins reacting with proteins other than HCAR1. This resulted in staining of perivascular pial cells in wild-type

but not in knockout mice, as now shown in the new Fig. 4a,b. Moreover, the HCARI immunoreactivity colocalized with mRFP-HCARI (new Fig. 4c,d,f). In view of this positive result, we have now deleted the previous Supplementary Table 1 and the corresponding text. We thank the Reviewer #3 for prompting us to do this experiment!

«- The rosiglitazone experiment is not particularly useful. The authors use this drug to elevate HCARI expression to determine where it is expressed. The drug, however, may cause a change in the expression of the receptor as opposed to just an increase in the cells that already express it.»

RESPONSE: *In view of this remark, and a similar remark by the Reviewer #2, we have now deleted the rosiglitazone data.*

Reviewer #4 (Remarks to the Author):

«This is an interesting paper that purports to show that the activation of lactate receptors at pial vessels results in cerebral angiogenesis. The authors provide data from diverse protocols to support their conclusions. In general the paper is well written and the data presented clearly. The claim that increased circulating lactate can induce angiogenesis is somewhat novel and certainly intriguing with far reaching therapeutic consequences if validated. There are, however, a number of critical issues that should be addressed.»

RESPONSE: *We appreciate that the Reviewer finds our paper to be interesting, well written and that the data are clearly presented, and also novel, intriguing and with potentially far reaching therapeutic consequences. We now address the critical issues brought up.*

«1. The most important issue concerns the methodology for determining angiogenesis, which underlies the major point of the paper. This methodology is not discussed in any detail. It appears to be based on a ratio of fluorescence positive (to Collagen IV antibody) pixels to tissue background pixels and presented as a volume percent.»

RESPONSE: *We apologize for previously failing to describe these methods clearly. We did NOT measure a ratio between positive and negative pixels. Instead, we inserted an array of points (using the ImageJ software with the SimpleGrid plugin) as an overlay on our images, and counted the percentage of points over vessels (versus number of points over the total area of interest) in a projection of a z-stack. The details are now included in the Methods section.*

«There are several concerns. A) Does this method distinguish changes in dilation?»

RESPONSE: *In the original version of the manuscript, we did not distinguish changes in dilatation from other changes. Based on the Reviewer's comment, we have now measured the average diameter of the capillaries in each animal. These data, presented in the legend of Fig. 1 and in the new Supplementary Fig. 4, show that there is no difference in dilatation between the groups. Therefore, we conclude that the measured difference in percent area covered by capillaries does not represent an increase in the calibre of the vessels.*

«B) Does the method account for regional and cortical layer differences in capillary density?»

RESPONSE: *As now described in the Methods, we sampled the whole of the hilus of the dentate gyrus, all layers of the sensorimotor cortex, and all the three layers of folia 1 and 2 of*

the cerebellar cortex, in parasagittal sections immunostained for collagen IV. Regional and layer differences in the cortex were therefore not examined.

«C) How were sampling errors minimized in such heterogeneous tissues?»

RESPONSE: Sampling errors in connection with this analysis were minimized by blinded collection of data such as point-counting for capillary density and measurement of capillary diameter, and by choosing sampling areas that are clearly defined. Thus the definition of the hilus region is easy and unambiguous, and so is the demarcation of the cerebellar and cerebral cortex against the underlying white matter.

«D) Were the data recorded by blinded investigators?»

RESPONSE: Yes, all analyses and collection of data such as point-counting for capillary density and measurement of capillary diameter were performed by an observer who was blinded to the genotypes and treatments. We apologize for having left out this important piece of information; it is now included in the description of the Methods.

«E) The volume percent reported (approx.. 8%) is about double that expected for average cerebral cortex (the hippocampal data are more in line with expectations).»

RESPONSE: As explained in responses to the Reviewer #1, the percentages of volume covered by vessels given in the previous version of the manuscript are not accurate absolute measures. The details about this method are now included in the Methods description. However, the literature data on capillary density vary, and are largely consistent with our data (please see references quoted under response to Reviewer #1).

Importantly, the purpose of the current study is not to determine the exact capillary density in the brain, but rather to investigate the effects of treatments on vascularization, for which the way we determined vascular density is valid. However, to avoid confusion, we have now normalized our data to the wild-type control group.

«F) It is not clear which vessels were included in the measurement (arterioles?, capillaries?, venules?).»

RESPONSE: Based on this comment, we have revisited our data, and removed any vessels larger than 10 μ m in diameter, so that only capillaries are included in the analysis. The same measure was taken when acquiring the new data from the sensorimotor cortex (revised Fig. 1a,b) and cerebellar cortex (Supplementary Fig. 4). This information is now included in the Methods description. The corrections did not affect the results obtained in hippocampus in the wild-type mice (revised Fig. 1c,d), nor the conclusion of the paper.

«2. The investigators follow up their initial findings with experiments designed to test potential molecular mechanisms. But, it is unclear how activation of pial associated cells would directly affect capillary endothelial cells at a distance. Even though changes may be detected, it is not possible to conclude that these were the only changes, or that a cause and effect can be shown.»

RESPONSE: New pictures are now shown, in the revised Fig. 3i, underlining the fact that HCARI carrying blood vessels extend from the pia mater all the way into the hippocampus, where the effects on VEGFA and angiogenesis are demonstrated. Likewise, HCARI carrying vessels supply all of the cerebral cortex. A likely scenario, indicated in the new Fig. 5 (previous Fig. 4), is that perivascular HCARI induces local production of VEGFA, which

then flows with the blood stream into the intrahippocampal and intracortical microvessels to cause their proliferation. However, we agree that the action might also take place through HCARI located elsewhere than on the vessels shown to have a high mRFP signal. We have therefore deleted the term “at pial vessels” from the title.

«3. There are no error bars provided in Supplemental figure 2.»

RESPONSE: We thank the Reviewer for spotting this oversight and have now added bars showing \pm s.d.

«4. The data shown in supplemental figure 3 do not show that the responses of the different groups were the same, only that no statistical differences were detected. It is likely that the test does not have the power to discriminate any differences. The authors gloss over this by using a larger than necessary y-scale scale and error bars based on s.d., whereas all other graphs use s.e.m.»

RESPONSE: We do not understand the rationale of this comment. The averages in the four individual experimental groups are the same (almost exactly 30%). The test used is a standard test for anxiety related behaviour (Shepherd JK et al 1994 Psychopharmacology). While we cannot rule out the possibility that a different type of test might reveal differences, this would not be related to the statistics or the discriminating power of the present test. The score in this test spans 0%-100%, which is why we give this range on the y-scale. We do not «gloss over» something by using s.d. rather than s.e.m.: $s.e.m. = s.d. / \sqrt{n}$, and all the values of n are specified in the legend. Furthermore, it is the statistics analysis (not the span of the y-scale, or whether s.e.m. or s.d. is shown in the figure) that measures whether there are significant differences. In this case $P = 0.957$ (one-way ANOVA), indicating, with 95.7% confidence, that the groups do not differ in the test.

«5. Claiming statistical significance to p equals 0.01 or smaller is not justified by the numbers of observations because reasonable test power cannot be maintained. (Button, K. S., et al. (2013). "Power failure: why small sample size undermines the reliability of neuroscience." Nat Rev Neurosci 14(5): 365-376.) $P < 0.05$ should be sufficient to make the point.»

RESPONSE: While there is some truth in this argument, the article referred to (Button KS et al. 2013 Nat Rev Neurosci (365)) was severely criticized for missing the real point (e.g., Bachetti P 2013 Nat Rev Neurosci “Small sample size is not the real problem”). Scientists are generally aware of the fact that a P -value is an estimate of likelihood, but no guarantee. The reader can make judgements taking also sample size and other information into consideration. We believe it is justified to adhere to the common usage of stating the P -values delivered by the statistical tests.

«6. In the analysis of the data from supplementary figure 8, how did the investigators control for changes in fiber volume, which is a well known mechanism for decreasing intercapillary distance and increase capillary density without angiogenesis?»

RESPONSE: We now present additional Supplementary Figs. 9i and j, which show that the volume (cross-sectional area) of the individual muscle fibres did not change significantly following exercise or lactate injections, in neither one of the two parts examined of the gastrocnemius muscle. This finding is in line with the sparse changes observed in fibre diameter in gastrocnemius muscles in endurance trained mice, including no change in type II fibres, the most abundant fibre type in gastrocnemius (Krüger K et al. 2013 PLoS One).

The authors have revised their manuscript. I am happy with the amendments regarding my concerns 1 and 2. However, issue 3 and 4 remain very critical and I am still very worried about the correctness of the data. I feel that the authors were not able to adequately address my concerns. I consider them as fundamental.

«3. The volume fraction (Figure 1) rises from 8% (which is already high, see above) to 12% which is really too high and this increase is extraordinary and very hard to believe.»

RESPONSE: *As described above, we agree that the absolute numbers given in the previous version of the manuscript may be overestimated, although similar values appear in the literature. However, the relative values are valid for comparison among the experimental groups. Indeed, the ca 40% increase in capillary area fraction reported here, is in line with the 49% increase (in ‘vascular surface area’, corresponding to the area fraction recorded here) that was reported by van Praag and coworkers after exercise (van Praag H, Shubert T, Zhao C & Gage FH 2005 J Neurosci; their Fig. 3b).*

The authors should have a look at the work of the Kleinfeld group. They will see that a volume fraction at this magnitude in cortex is simply not realistic.

Blinder et al. Nat Neurosci. 2013; 16(7):889-97.

Tsai et al. J Neurosci. 2009; 29(46):14553-70.

The authors have chosen to give relative values instead, that’s fine. However, I am still worried that the underlying data is solid. What worries me most is the fact that Figure 1 a and b completely changed. From going from absolute (original submission) to relative (revision) values, the effects of exercise/lactate and genetic manipulation must stay the same, unless the data were changed. This is the most important figure in the paper and now all of a sudden the effects have shrunk down significantly!

Figure 1b from original submission (left) and revised figure (right). The data is now given in relative values but the underlying data clearly changed! Why?

«4. Blood plasma lactate levels are crucial for this study. The supplemental figure 1 shows the plasma lactate kinetics after the subcutaneous shots of lactate versus PBS. However, the lactate levels are way too high. The PBS treated mice show a lactate level of higher than 5 after 180 minutes! This is about 5-times too high. Something is clearly wrong here, as plasma levels of around 1 mM are expected. These data, when re-visited, belong in the main part of the paper.»

RESPONSE: *We are not aware of which published articles the Reviewer uses as basis for suggesting that our values are too high and that a value of 1 mM of plasma lactate is expected. Perhaps the Reviewer has in mind fasting levels of lactate? These are lower than fed values. Thus Wang MW et al 1992 Endocr Res found basal lactate values of 2.16 ± 0.18 mM and 3.10 ± 0.29 mM in fasted and fed mice, respectively. It should be noted that our mice were not fasted. Further, Iversen NK et al 2012 Respir Physiol Neurobiol ('The normal acid-base status of mice') took small blood samples from undisturbed mice with indwelling catheters in the carotid artery, finding lactate concentrations of 4.6 ± 0.7 mM. This is in excellent agreement with the values in our Supplementary Fig. 2. Similarly, Suhara T et al 2015 PNAS reported 3.7 mM (their Fig. 1B), and Flynn JM et al 2010 PLoS One reported 3.2 mM in wild-type control mice. Our control lactate value of 3.05 ± 0.72 mM (Supplementary Fig. 2) is therefore fully in agreement with literature data.*

We still think the figure in question belongs in the Supplementary material, because it presents background information rather than novel results, but leave the decision on this matter to the Editor's discretion.

I am not sure I understand the logic behind the statement that their 5 mM plasma lactate level is similar to the cited 3.1 mM. I am concerned with the PBS treatment that leads apparently to 5 mM plasma lactate (Suppl. Fig.). The lactate data is really key for this study and should not be hidden in the suppl. information.

Iversen's data was measured in BALBc mice, and a direct comparison is not necessarily possible. C57BL/6 plasma lactate levels in literature are:

2 mmol/l; Acharya et al, Scientific Reports, 2014
1.97 mmol/l; Graham et al., Nature, 1997
 1.75 ± 0.36 mmol/l; Hatchell and MacInnes, Genetics, 1973
 1.7 ± 0.2 mmol/l; Zykova et al., Diabetes, 2000

«5. The authors make use of in vivo two-photon microscopy to show the localization of mRFP-HCAR1. It would be really important to corroborate the vascular density data (see my point 3) with in vivo two-photon stacks of a fluorescent dextran conjugate filled vessels. I guess these data are available anyways and would make a strong case in confirming the histological vascular density analysis, which in my view is not yet done well (see my point 2).»

RESPONSE: *The trained and lactate injected mice were not examined by two-photon microscopy, and it does not seem reasonable to repeat this experiment with mice that have cranial windows for two-photon examination. As explained above, it is the relative change in vascular volume that is important for the conclusions of the paper, not the absolute values.*

However, there are data in the literature on capillary volume examined by two-photon microscopy in vivo. Above we already quoted the paper by Desjardins M et al 2014 Neurobiol Aging, who used two-photon microscopy of intravascular fluorescent dextran in live young rats, and estimated the capillary volume fraction in the sensorimotor cortex at 6.8 ± 0.3 %, and the internal capillary diameter at 6.2 ± 0.1 μm . These values are in agreement with our data in mouse hippocampus (7% and 6 μm , respectively).

The study by Desjardins et al was performed in Long Evans rats, not in mice, so the data are not comparable!

Reviewer #2 (Remarks to the Author):

The authors have improved the manuscript with their careful revisions in response to the reviewers' comments.

Reviewer #3 (Remarks to the Author):

In the resubmission Morland et al. addressed most of the concerns with the manuscript. They now present a compelling finding that lactate released after intense exercising induces cerebral angiogenesis via the lactate receptor (HCAR1). This is a very interesting finding that has important implications for aging.

In the first submission the main concern was that the authors claimed that the key cell type expressing the HCAR1 was pial vessels, however they failed to show that HCAR1 in these cells was how the signal was transduced. The authors modified the manuscript to remove the claim that the pial vessels were the key cell type. They removed this claim in the title, abstract and throughout the paper. Therefore the paper now doesn't present any non-supported conclusions, but still presents a significant step forward. The one place that the authors still need to change their conclusions is in the model presented in figure 5. This model still claims that pial vessels are the key cells transmitting the lactate signal. They should either remove this model, change it to not include the pial vessels, or alter it to make sure it is clear that the role of the pial vessels is still uncertain.

The authors also addressed all of the minor concerns.

Reviewer #4 (Remarks to the Author):

No further comments

Response to Reviewers, revised manuscript NCOMMS-16-15634B

Reviewer #1, point 3 (Referee_1_report_1487284544_1.pdf):

«The authors have chosen to give relative values instead, that's fine. However, I am still worried that the underlying data is solid. What worries me most is the fact that Figure 1 a and b completely changed. From going from absolute (original submission) to relative (revision) values, the effects of exercise/lactate and genetic manipulation must stay the same, unless the data were changed. This is the most important figure in the paper and now all of a sudden the effects have shrunk down significantly!»

RESPONSE: It appears that the chief concern of the Reviewer #1 arose because this Reviewer did not realize that our revised Fig. 1a,b is based on an entirely new dataset. (We see now that this could have been pointed out more explicitly in our rebuttal letter.)

As described in the revised Methods, and in our previous Response to Editors and Reviewers (Reviewer #1, point 2, and Reviewer #4, point 1F), the new Fig. 1a,b presents new information. Here, the entire rostro-caudal stretch (ca 4 mm) of the sensorimotor cortex gray matter in parasagittal sections was analyzed (by blind point-counting) using images generated by an automated slide scanner system. This gives a much larger set of observations than the previous confocal images from a restricted part of the M1 motor cortex, with less variation, and represents a much larger part of the cerebral cortex.

Another reason prompting us to perform the new analyses was to include only capillaries, defined as vessels with external diameter of 10 μm or less (Hall CN et al 2014 Nature). This followed from the requirement by the Reviewer #4 to define the type of vessels included in the analyses. [For the circumscribed hilus region of hippocampus (Fig. 1c,d), where the higher resolution of confocal microscopy offers an advantage, while the ability to sample large areas by the scanner system offers no advantage, the few vessels exceeding 10 μm in diameter were excluded from the dataset and the values corrected correspondingly.]

The new results (for extensive areas of the sensorimotor cortex) show a lesser percentage change than the previous (for a restricted area of motor cortex), but are at least as robust with respect to statistical significance. The same type of analysis in the cerebellar cortex, in the same collection of sections from the same mice, showed no significant changes (new Supplementary Fig. 4a,b).

We have now clarified the description in the Methods concerning brain capillary density. In sum, the new results strengthen the validity of our findings on capillary density.

«The authors should have a look at the work of the Kleinfeld group. They will see that a volume fraction at this magnitude in cortex is simply not realistic.

Blinder et al. Nat Neurosci. 2013; 16(7):889-97.

Tsai et al. J Neurosci. 2009; 29(46):14553-70.»

RESPONSE: [«a volume fraction at this magnitude» refers to a figure of 8%.] We do agree with the Reviewer #1 that 8% is an overestimate of the capillary volume fraction (although there are several publications in the literature presenting such high values, quoted in our previous rebuttal letter). However, the primary observations are really of area fraction as a

proxy for volume fraction. As shown below, the values in control mice can be converted to a volume fraction of about 2%, but the correction cancels out when control and experiment are compared to determine % change, which is the important point for the conclusion of our paper.

New analysis for Fig. 1b, % area fraction of capillaries in sensorimotor cortex, mean±s.d.:

wt vehicle	wt EX	wt Lact	ko vehicle	ko EX	ko Lact
8.36±0.49	9.40±0.45	9.99±0.86	8.44±0.31	8.13±0.68	8.31±0.69

(These data are now shown as the new Supplementary Table 2, together with the calculation presented below.)

Taking volume fraction (VF) as equal to area fraction (AF), according to the Delesse principle, is strictly valid only for infinitely thin sections (of thickness $t = 0$, like on the polished cut face of a rock), but for sections of finite thickness, the VF is still proportional to AF. According to a reference cited by the Reviewer #1 (Weber B et al 2008 Cereb Cortex), in its Appendix on stereological computations (p 2328), the volume fraction of capillaries (VF) can be calculated from their mean diameter (dV), their area fraction (AF) and the section thickness (t) as $VF = dV * AF / t$. We have determined dV to be about 5.8 μm (outer diameter, legend of new Fig.1). In the new data for sensorimotor cortex, AF = 8.4% in wild-type as well as knockout controls (table above), and the section thickness $t = 20 \mu\text{m}$. This gives $VF = 5.8\mu\text{m} * 8.4\% / 20\mu\text{m} = \underline{2.4\%}$.

With the wisdom of hindsight, we should have presented this corrected volume fraction in the revision of our paper, for the benefit of Reviewer #1. However, the main point is that only the % changes caused by the interventions are important for the conclusion of the paper. Nevertheless, we now include in the paper the observed AF values and the correction for capillary diameter and section thickness to obtain VF (new Supplementary Table 2).

The corrected basal value of 2.4% capillary volume fraction in the sensorimotor cortex agrees well with the literature. For example, five different publications (Verant P et al 2007 J Cereb Blood Flow Metab, Serduc R et al 2006 Int J Radiat Oncol Biol Phys, Zhao R, Pollack GM 2007 Biochem Pharmacol, Heintzer S et al 2006 Neuroimage, Heintzer S et al 2008 Neuroimage) give values for cerebrocortical microvasculature in mice ranging from 1.5% to 3.6% (median 2.5%). Notably, Verant P et al 2007 J Cereb Blood Flow Metab determined the cortical capillary blood volume fraction at 2.0% - 2.4% by two-photon microscopy in live anesthetized mice.

The Reviewer #1 cites two references on volume fraction [of microvessels]. The first one (Blinder et al 2013 Nat Neurosci) does not actually determine volume fraction, but refers to the second (Tsai PS et al 2009 J Neurosci), which reports values based on “vessel enhanced grayscale volume” determined automatically in brain sections. This paper reports (their Fig. 13 top) fractional vascular volumes for microvessels (diameter $< 6\mu\text{m}$) in mouse cerebral cortex at 0.6% in C57B6 mice and 0.7% in Swiss mice. These values are somewhat lower than most literature data on mice, as exemplified above. (In this work by Tsai et al, the mice were perfused transcardially with a large volume of PBS (60ml), preceding the fixative (60ml 4% PFA), then gelatin/fluorescent albumin (20ml) at 40°C, with the mice submerged in ice. It may well be asked whether this treatment caused changes in the diameter and state of the

blood vessels. It may also be asked how well the "vessel enhanced grayscale volume" represented the real volume. Be this as it may, their lower cut-off value for including vessels, <6µm versus <10µm, implies a lower estimated volume fraction compared to ours.)

Based on the facts listed above, we do hope the Reviewer and Editors would agree that our findings of increased capillary density after exercise and lactate injections are sound and firmly based.

Reviewer #1, point 4 (Referee_1_report_1487284544_1.pdf):

«I am not sure I understand the logic behind the statement that their 5 mM plasma lactate level is similar to the cited 3.1 mM.»

RESPONSE: We cannot find any such statement, neither in our manuscript, nor in the rebuttal letter. The basal plasma lactate concentration (without injection) presented in the Supplementary Fig. 2, is 3.05 ± 0.72 mM (mean \pm s.d., n = 4).

[Incidentally, we noticed that in the previous Supplementary Fig. 2, 'serum' was erroneously stated instead of 'plasma', whereas the Methods show clearly that plasma was measured. This error is corrected in the revised Supplementary Fig. 2.]

«I am concerned with the PBS treatment that leads apparently to 5 mM plasma lactate (Suppl. Fig.).»

RESPONSE: The final time point in control mice cannot alone be taken as representative of effects of PBS injections. For the controls, each time point is based on only three mice. Out of the five time-points determined for PBS injections in Supplementary Fig. 2, three are at about 3 mM, the median value of all five is 3.3 mM, i.e., essentially equal to the basal concentration of 3.05 mM. Moreover, Chi-square analysis of the mean lactate values at the different time points after PBS injections, compared with the basal value shows no significant difference, $P = 0.91$. (A similar comparison of the values after lactate injections with the basal value shows a highly significant difference, $P \ll 0.001$.) Thus our data cannot be interpreted as indicating that plasma lactate is increased after saline injections. A likely cause of variations is in the procedure of obtaining samples, which comprises variations such as in the light isoflurane anesthesia and stress before anesthesia and decapitation. Indeed, volatile anesthetics are known to greatly increase blood lactate levels (Horn T, Klein J 2010 Neurochem Int), an effect that must be expected to vary, particularly when light anesthesia is aimed for, as was the case here. The handling of the mice to obtain the blood samples could well cause some disturbance of the basal plasma levels of lactate, but this would be of no consequence to the condition of the mice participating in the experiments shown in the main text and Figs of the paper, because these mice were not exposed to such handling.

«The lactate data is really key for this study and should not be hidden in the suppl. information.»

RESPONSE: The Supplementary Fig. 2 is a pilot experiment that simply serves to show the time-course and the approximate magnitude of the rise in plasma lactate after subcutaneous injection. It is important to know that the subcutaneous lactate injections caused a peak of plasma lactate at about 5 - 15 min, reaching above 10 mM, and a significant elevation of lactate for 30 min. However, this knowledge is not influenced by minor differences in the lactate levels in control mice and whether there may be some imprecision in in the

measurements. The observations made agree with those of E L et al 2013 J Neurochem who also reported a peak of plasma lactate at 5 - 15 min, after intraperitoneal injection. We think the time-course of plasma lactate represents background information rather than novel results, and therefore belongs in the Supplementary material. (But of course, as already stated, the decision on this matter is up to the Editor's discretion.)

«Iversen's data was measured in BALBc mice, and a direct comparison is not necessarily possible.»

RESPONSE: The Reviewer is of course right in this statement. However, we hope the Reviewer and the Editors will agree that information on in vivo values in undisturbed mice is relevant in the context, irrespective of the strain.

«C57BL/6 plasma lactate levels in literature are:
2 mmol/l; Acharya et al, Scientific Reports, 2014
1.97 mmol/l; Graham et al., Nature, 1997
1.75 ± 0.36 mmol/l; Hatchell and MacInnes, Genetics, 1973
1.7 ± 0.2 mmol/l; Zykova et al., Diabetes, 2000»

RESPONSE: At least one of the papers listed by the Reviewer (Zykova SN et al 2000 Diabetes) used fasted mice, while the others do not specify whether the mice were fed or fasted. As our mice were not fasted, we expect somewhat higher values than those of Zykova et al. In our Response to Editors and Reviewers, we already cited Wang MW et al 1992 Endocr Res who found basal lactate levels of 2.16 ± 0.18 mM in fasted and 3.10 ± 0.29 mM in fed mice, respectively. This corresponds rather closely to the difference between the results of Zykova et al for fasted mice (about 2 mM) and ours for fed control mice (about 3 mM). Our control values also agree with other papers than those listed by the Reviewer, for example Flynn JM et al 2010 PLoS One (their Fig. 2A) reported 3.2 mM in wild-type control C57/BL/6 mice.

In all, it is not reasonable to doubt our findings, based on the reported levels of lactate.

Reviewer #1, point 5 (Referee_1_report_1487284544_1.pdf):

«The study by Desjardins et al was performed in Long Evans rats, not in mice, so the data are not comparable!»

RESPONSE: The Reviewer is of course right here. It appears from the literature, though, that brain fractional vascular volume varies no more between species than between different studies in the same species (see, e.g., survey by Tsai PS et al 2009 J Neurosci, their Fig. 13).

Reviewer #3 (Remarks to the Author):

«In the resubmission Morland et al. addressed most of the concerns with the manuscript. They now present a compelling finding that lactate released after intense exercising induces cerebral angiogenesis via the lactate receptor (HCAR1). This is a very interesting finding that has important implications for aging.

In the first submission the main concern was that the authors claimed that the key cell type expressing the HCAR1 was pial vessels, however they failed to show that HCAR1 in these cells was how the signal was transduced. The authors modified the manuscript to remove the claim that the pial vessels were the key cell type. They removed this claim in the title, abstract and throughout the paper. Therefore the paper now doesn't present any non-

supported conclusions, but still presents a significant step forward. The one place that the authors still need to change their conclusions is in the model presented in figure 5. This model still claims that pial vessels are the key cells transmitting the lactate signal. They should either remove this model, change it to not include the pial vessels, or alter it to make sure it is clear that the role of the pial vessels is still uncertain.»

RESPONSE: We are happy for these supportive comments. Concerning the model presented in Fig. 5, we believe the third alternative suggested by the Reviewer #3 is the best, «alter it to make sure it is clear that the role of the pial vessels is still uncertain». In the revised version of Fig. 5, we have therefore put a question mark on the arrows connecting HCAR1 expressing cells and VEGFA, and correspondingly in the legend: The legend emphasizes the uncertainty of the role of the pial vessels: “.. inset indicates possible, yet unidentified (?), pathways ..”, and subsequently outlines possibilities. In addition, we have removed “leptomeningeal” from the figure itself and from the Fig. 5 caption, in view of the facts that 1) this designation does not strictly fit for the cells of the intracerebral perivascular sheaths, 2) the latter HCAR1 carrying cells share some properties with pericytes, and 3) other cells could carry HCAR1 at levels subthreshold for microscopic detection, but still cause elevation of VEGFA. We hope these alterations take care of the final point raised by the Reviewer #3.

Response to Reviewers, revised manuscript NCOMMS-16-15634B

Reviewer #1, point 3 (Referee_1_report_1487284544_1.pdf):

«The authors have chosen to give relative values instead, that's fine. However, I am still worried that the underlying data is solid. What worries me most is the fact that Figure 1 a and b completely changed. From going from absolute (original submission) to relative (revision) values, the effects of exercise/lactate and genetic manipulation must stay the same, unless the data were changed. This is the most important figure in the paper and now all of a sudden the effects have shrunk down significantly!»

RESPONSE: It appears that the chief concern of the Reviewer #1 arose because this Reviewer did not realize that our revised Fig. 1a,b is based on an entirely new dataset. (We see now that this could have been pointed out more explicitly in our rebuttal letter.)

As described in the revised Methods, and in our previous Response to Editors and Reviewers (Reviewer #1, point 2, and Reviewer #4, point 1F), the new Fig. 1a,b presents new information. Here, the entire rostral-caudal stretch (ca 4 mm) of the sensorimotor cortex gray matter in parasagittal sections was analyzed (by blind point-counting) using images generated by an automated slide scanner system. This gives a much larger set of observations than the previous confocal images from a restricted part of the M1 motor cortex, with less variation, and represents a much larger part of the cerebral cortex.

Another reason prompting us to perform the new analyses was to include only capillaries, defined as vessels with external diameter of 10 μm or less (Hall CN et al 2014 Nature). This followed from the requirement by the Reviewer #4 to define the type of vessels included in the analyses. [For the circumscribed hilus region of hippocampus (Fig. 1c,d), where the higher resolution of confocal microscopy offers an advantage, while the ability to sample large areas by the scanner system offers no advantage, the few vessels exceeding 10 μm in diameter were excluded from the dataset and the values corrected correspondingly.]

The new results (for extensive areas of the sensorimotor cortex) show a lesser percentage change than the previous (for a restricted area of motor cortex), but are at least as robust with respect to statistical significance. The same type of analysis in the cerebellar cortex, in the same collection of sections from the same mice, showed no significant changes (new Supplementary Fig. 4a,b).

We have now clarified the description in the Methods concerning brain capillary density. In sum, the new results strengthen the validity of our findings on capillary density.

Indeed, it would have been important to inform the reviewers that an entirely new data set was used. Also, I am still concerned that by enlarging the sample size the effect size went down so dramatically. Increasing the sample size decreases the variance but not the mean, unless there is something hidden in the data that is not understood.

«The authors should have a look at the work of the Kleinfeld group. They will see that a volume fraction at this magnitude in cortex is simply not realistic.

Blinder et al. Nat Neurosci. 2013; 16(7):889-97.

Tsai et al. J Neurosci. 2009; 29(46):14553-70.»

RESPONSE: [«a volume fraction at this magnitude» refers to a figure of 8%.] We do agree with the Reviewer #1 that 8% is an overestimate of the capillary volume fraction (although there are several publications in the literature presenting such high values, quoted in our previous rebuttal letter). However, the primary observations are really of area fraction as a proxy for volume fraction. As shown below, the values in control mice can be converted to a volume fraction of about 2%, but the correction cancels out when control and experiment are compared to determine % change, which is the important point for the conclusion of our paper.

New analysis for Fig. 1b, % area fraction of capillaries in sensorimotor cortex, mean±s.d.:

wt vehicle	wt EX	wt Lact	ko vehicle	ko EX	ko Lact
8.36±0.49	9.40±0.45	9.99±0.86	8.44±0.31	8.13±0.68	8.31±0.69

(These data are now shown as the new Supplementary Table 2, together with the calculation presented below.)

Taking volume fraction (VF) as equal to area fraction (AF), according to the Delesse principle, is strictly valid only for infinitely thin sections (of thickness $t = 0$, like on the polished cut face of a rock), but for sections of finite thickness, the VF is still proportional to AF. According to a reference cited by the Reviewer #1 (Weber B et al 2008 Cereb Cortex), in its Appendix on stereological computations (p 2328), the volume fraction of capillaries (VF) can be calculated from their mean diameter (dV), their area fraction (AF) and the section thickness (t) as $VF = dV \cdot AF / t$. We have determined dV to be about 5.8 μm (outer diameter, legend of new Fig.1). In the new data for sensorimotor cortex, AF = 8.4% in wild-type as well as knockout controls (table above), and the section thickness $t = 20 \mu\text{m}$. This gives $VF = 5.8\mu\text{m} \cdot 8.4\% / 20\mu\text{m} = \underline{2.4\%}$.

With the wisdom of hindsight, we should have presented this corrected volume fraction in the revision of our paper, for the benefit of Reviewer #1. However, the main point is that only the % changes caused by the interventions are important for the conclusion of the paper. Nevertheless, we now include in the paper the observed AF values and the correction for capillary diameter and section thickness to obtain VF (new Supplementary Table 2).

The corrected basal value of 2.4% capillary volume fraction in the sensorimotor cortex agrees well with the literature. For example, five different publications (Verant P et al 2007 J Cereb Blood Flow Metab, Serduc R et al 2006 Int J Radiat Oncol Biol Phys, Zhao R, Pollack GM 2007 Biochem Pharmacol, Heintzer S et al 2006 Neuroimage, Heintzer S et al 2008 Neuroimage) give values for cerebrocortical microvasculature in mice ranging from 1.5% to 3.6% (median 2.5%). Notably, Verant P et al 2007 J Cereb Blood Flow Metab determined the cortical capillary blood volume fraction at 2.0% - 2.4% by two-photon microscopy in live anesthetized mice.

The Reviewer #1 cites two references on volume fraction [of microvessels]. The first one (Blinder et al 2013 Nat Neurosci) does not actually determine volume fraction, but refers to the second (Tsai PS et al 2009 J Neurosci), which reports values based on “vessel enhanced grayscale volume” determined automatically in brain sections. This paper reports (their Fig.

13 top) fractional vascular volumes for microvessels (diameter $<6\mu\text{m}$) in mouse cerebral cortex at 0.6% in C57B6 mice and 0.7% in Swiss mice. These values are somewhat lower than most literature data on mice, as exemplified above. (In this work by Tsai et al, the mice were perfused transcardially with a large volume of PBS (60ml), preceding the fixative (60ml 4% PFA), then gelatin/fluorescent albumin (20ml) at 40°C, with the mice submerged in ice. It may well be asked whether this treatment caused changes in the diameter and state of the blood vessels. It may also be asked how well the "vessel enhanced grayscale volume" represented the real volume. Be this as it may, their lower cut-off value for including vessels, $<6\mu\text{m}$ versus $<10\mu\text{m}$, implies a lower estimated volume fraction compared to ours.)

Based on the facts listed above, we do hope the Reviewer and Editors would agree that our findings of increased capillary density after exercise and lactate injections are sound and firmly based.

I am not impressed by the reasoning. The perfusion protocol in Tsai has been thoroughly developed to minimize vessel deformation and scaling. Also, the diameter values were always compared with in vivo two-photon microscopy. Furthermore, the capillary cutoff of 10 microns is way too high, since in a 10-micron vessel, erythrocytes do not exhibit single file flow. 6 microns is clearly more realistic, no matter what other groups who have no or very little in vivo experience use (eg. Hall et al.).

Reviewer #1, point 4 (Referee_1_report_1487284544_1.pdf):

«I am not sure I understand the logic behind the statement that their 5 mM plasma lactate level is similar to the cited 3.1 mM.»

RESPONSE: We cannot find any such statement, neither in our manuscript, nor in the rebuttal letter. The basal plasma lactate concentration (without injection) presented in the Supplementary Fig. 2, is 3.05 ± 0.72 mM (mean \pm s.d., $n = 4$).

The authors show 5 mM lactate level after 180 min (Suppl. Figure 2) and state that their values are in line with the literature. This is my point. I agree that there was no explicit statement. But implicitly, this is what the authors claim.

«I am concerned with the PBS treatment that leads apparently to 5 mM plasma lactate (Suppl. Fig.).»

RESPONSE: The final time point in control mice cannot alone be taken as representative of effects of PBS injections. For the controls, each time point is based on only three mice. Out of the five time-points determined for PBS injections in Supplementary Fig. 2, three are at about 3 mM, the median value of all five is 3.3 mM, i.e., essentially equal to the basal concentration of 3.05 mM. Moreover, Chi-square analysis of the mean lactate values at the different time points after PBS injections, compared with the basal value shows no significant difference, $P = 0.91$. (A similar comparison of the values after lactate injections with the basal value shows a highly significant difference, $P \ll 0.001$.) Thus our data cannot be interpreted as indicating that plasma lactate is increased after saline injections. A likely cause of variations is in the procedure of obtaining samples, which comprises variations such as in the light isoflurane anesthesia and stress before anesthesia and decapitation. Indeed, volatile anesthetics are known to greatly increase blood lactate levels (Horn T, Klein J 2010

Neurochem Int), an effect that must be expected to vary, particularly when light anesthesia is aimed for, as was the case here. The handling of the mice to obtain the blood samples could well cause some disturbance of the basal plasma levels of lactate, but this would be of no consequence to the condition of the mice participating in the experiments shown in the main text and Figs of the paper, because these mice were not exposed to such handling.

I do not agree. An effect size of this magnitude (from 3.05 to 5 mM) cannot be just waved away. More animals are required and then show that this 5 mM has been created by outliers. It is as simple as that.

It is also not understandable that the control mice were treated differently (2.5 versus 2 g / kg).

«The lactate data is really key for this study and should not be hidden in the suppl. information.»

RESPONSE: The Supplementary Fig. 2 is a pilot experiment that simply serves to show the time-course and the approximate magnitude of the rise in plasma lactate after subcutaneous injection. It is important to know that the subcutaneous lactate injections caused a peak of plasma lactate at about 5 - 15 min, reaching above 10 mM, and a significant elevation of lactate for 30 min. However, this knowledge is not influenced by minor differences in the lactate levels in control mice and whether there may be some imprecision in the measurements. The observations made agree with those of E L et al 2013 J Neurochem who also reported a peak of plasma lactate at 5 - 15 min, after intraperitoneal injection. We think the time-course of plasma lactate represents background information rather than novel results, and therefore belongs in the Supplementary material. (But of course, as already stated, the decision on this matter is up to the Editor's discretion.)

«Iversen's data was measured in BALBc mice, and a direct comparison is not necessarily possible.»

RESPONSE: The Reviewer is of course right in this statement. However, we hope the Reviewer and the Editors will agree that information on in vivo values in undisturbed mice is relevant in the context, irrespective of the strain.

«C57BL/6 plasma lactate levels in literature are:

2 mmol/l; Acharya et al, Scientific Reports, 2014

1.97 mmol/l; Graham et al., Nature, 1997

1.75 ± 0.36 mmol/l; Hatchell and MacInnes, Genetics, 1973

1.7 ± 0.2 mmol/l; Zykova et al., Diabetes, 2000»

RESPONSE: At least one of the papers listed by the Reviewer (Zykova SN et al 2000 Diabetes) used fasted mice, while the others do not specify whether the mice were fed or fasted. As our mice were not fasted, we expect somewhat higher values than those of Zykova et al. In our Response to Editors and Reviewers, we already cited Wang MW et al 1992 Endocr Res who found basal lactate levels of 2.16 ± 0.18 mM in fasted and 3.10 ± 0.29 mM in fed mice, respectively. This corresponds rather closely to the difference between the results of Zykova et al for fasted mice (about 2 mM) and ours for fed control mice (about 3 mM). Our control values also agree with other papers than those listed by the Reviewer, for example Flynn JM et al 2010 PLoS One (their Fig. 2A) reported 3.2 mM in wild-type control C57/BL/6 mice.

In all, it is not reasonable to doubt our findings, based on the reported levels of lactate.

Here, I cannot say more at this point. I cannot follow the authors reasoning, and I am still convinced that the reported lactate levels (1) are too high and (2) belong in the main part of the figure and not in the supplemental information.

Reviewer #1, point 5 (Referee_1_report_1487284544_1.pdf):

«The study by Desjardins et al was performed in Long Evans rats, not in mice, so the data are not comparable!»

RESPONSE: The Reviewer is of course right here. It appears from the literature, though, that brain fractional vascular volume varies no more between species than between different studies in the same species (see, e.g., survey by Tsai PS et al 2009 J Neurosci, their Fig. 13).

That's ok, errors can happen. However, "brain fractional vascular volume varies no more between species than between different studies in the same species" is a bold shortcut and lacks a proper study or meta-analysis.

RESPONSES to the last set of comments from the Reviewer#1:

«Indeed, it would have been important to inform the reviewers that an entirely new data set was used. Also, I am still concerned that by enlarging the sample size the effect size went down so dramatically. Increasing the sample size decreases the variance but not the mean, unless there is something hidden in the data that is not understood.»

RESPONSE: We agree on the first sentence, and apologize for having been insufficiently explicit about this. Concerning the effect of enlarging the sample size, this is only part of the point. As we state, a much larger area of the cortex and therefore a much more representative sample was analysed for the revised paper. While we do not exclude the possibility that future analyses could reveal regional differences in the effect, the present data are sufficient to make the important conclusion: in the cerebral cortex, there is an increase in capillary density induced by exercise and lactate through the lactate receptor.

«I am not impressed by the reasoning. The perfusion protocol in Tsai has been thoroughly developed to minimize vessel deformation and scaling. Also, the diameter values were always compared with in vivo two-photon microscopy. Furthermore, the capillary cutoff of 10 microns is way too high, since in a 10-micron vessel, erythrocytes do not exhibit single file flow. 6 microns is clearly more realistic, no matter what other groups who have no or very little in vivo experience use (eg. Hall et al.).»

RESPONSE: This item is not essential now, since correction for section thickness puts our basal values for capillary volume fraction (2.4%) in the middle of the range reported in the literature. Our point here was that the values given in the Tsai paper are lower than most data in the literature, a fact that we document by citations, including ones on in vivo two-photon microscopy. The Reviewer may be right that 6 micrometer could be a more suitable cutoff value for capillaries than 10 micrometer, particularly in studies focusing on capillary properties. In our study, the cutoff at 10 micrometer was taken from an authoritative study published in Nature and used as a practical definition of 'which vessels were included in the measurement' (requested by Reviewer #4).

«The authors show 5 mM lactate level after 180 min (Suppl. Figure 2) and state that their values are in line with the literature. This is my point. I agree that there was no explicit statement. But implicitly, this is what the authors claim.»

RESPONSE: No, we make no claim based the single data-point at 5 mM.

«I do not agree. An effect size of this magnitude (from 3.05 to 5 mM) cannot be just waved away. More animals are required and then show that this 5 mM has been created by outliers. It is as simple as that.

It is also not understandable that the control mice were treated differently (2.5 versus 2 g/kg).»

RESPONSE: The Reviewer #1 is right about what would be required, if the purpose were to establish the exact time course of blood lactate after injection, or whether the lactate levels in saline injected controls really differ from basal levels. This was not the purpose. The pilot experiment shown as supplemental information suffices to establish that subcutaneous lactate injections caused a peak of plasma lactate at about 5 - 15 min, reaching above 10 mM, and a significant elevation of lactate for 30 min. More accurate information on these points would not change the conclusions of our paper.

«That's ok, errors can happen. However, "brain fractional vascular volume varies no more between species than between different studies in the same species" is a bold shortcut and lacks a proper study or meta-analysis.»

RESPONSE: Again, the Reviewer #1 is ultimately right: a 'proper study and meta-analysis' would be required to rigorously establish the statement quoted. In the context, we think it was adequate with the reservation made ("It appears from the literature, though, that brain fractional vascular ..") and the reference given (to a paper reviewing data from multiple publications).